# Lost in Transmission:
# When and Why LLMs Fail to Reason Globally

**Tobias Schnabel**[*]  **Kiran Tomlinson**[*]  **Adith Swaminathan**  **Jennifer Neville**
Microsoft Research  Microsoft Research  Netflix  Microsoft Research

## Abstract

Despite their many successes, transformer-based large language models (LLMs) continue to struggle with tasks that require complex reasoning over large parts of their input. We argue that these failures arise due to capacity limits on the accurate flow of information within LLMs. To formalize this issue, we introduce the bounded attention prefix oracle (BAPO) model, a new computational framework that models bandwidth constraints on attention heads, the mechanism for internal communication in LLMs. We show that several important reasoning problems like graph reachability require high communication bandwidth for BAPOs to solve; we call these problems BAPO-hard. Our experiments corroborate our theoretical predictions: GPT-4o, Claude, and Gemini succeed on BAPO-easy tasks and fail even on relatively small BAPO-hard tasks. BAPOs also reveal another benefit of chain of thought (CoT): we prove that breaking down a task using CoT can turn any BAPO-hard problem into a BAPO-easy one. Our results offer principled explanations for key LLM failures and suggest directions for architectures and inference methods that mitigate bandwidth limits.

## 1 Introduction

Despite the empirical successes of transformer-based large language models (LLMs), they exhibit persistent failures on *global problems* that require integrating information across the entire input, such as chaining syllogisms [1], function composition [10], and formal language recognition [5]. Our core hypothesis is that these failures arise due to an inability of LLMs to accurately communicate information across *residual streams* [13], the sequences of transformer blocks corresponding to each input token. For an LLM to solve a problem, information about early tokens must be transmitted through attention into the last token's residual stream. While some problems, such as needle-in-a-haystack tasks, do not require much information to cross residual streams, we show that global problems with complex inter-token dependencies require substantial communication. We conjecture that LLMs fail on such problems due to a limit on the amount of information they can accurately transmit between residual streams, which we refer to as their *effective bandwidth*. This is supported by prior work on capacity bounds on attention [12] and sparse attention [9, 4, 33, 41]. Causal attention exacerbates communication issues by forcing pre-processing in early streams to be independent of later tokens, increasing the representation size of problems. Figure 1 illustrates the issue of information flow and previews our empirical results.

To formally analyze these issues, we propose the *bounded attention prefix oracle* (BAPO) model—a new computational model designed to capture the communication constraints and causal attention of transformer-based LLMs. BAPOs capture how much information must be communicated between residual streams to solve a problem, abstracting away other details of the transformer architecture. When producing an output, there are two ways in which transformers can integrate information about the last token with information from earlier tokens (i.e., from a prefix of the input): they can attend to

---

[*]Equal contribution.

39th Conference on Neural Information Processing Systems (NeurIPS 2025).

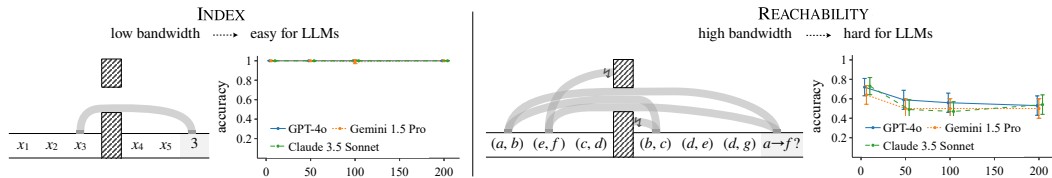

Figure 1: We conjecture that LLMs have a limit on their effective bandwidth, which we illustrate above by constrained information flow across one particular prefix–suffix split of the input. The BAPO model quantifies the communication bandwidth needed for transformers with causal attention to solve a problem. INDEX requires low communication bandwidth and LLMs solve it without issue; REACHABILITY requires high bandwidth and LLMs struggle with it.

precomputed values from higher layers of prefix streams, or they can attend to the raw prefix token values (or both; see Figure 2a). The BAPO model captures these two types of information flow and quantifies limits on the amount of information transmitted. In contrast with existing theoretical work on the expressivity of the transformer model class [40, 23, 15, 37, 34, 6], we seek to characterize the (in)ability of LLMs to solve global reasoning problems in practice. Strikingly, our experiments suggest that the effective bandwidth of modern LLMs is a small constant, so we call problems *BAPO-hard* if they cannot be solved by a constant-bandwidth BAPO and *BAPO-easy* if they can.

**Theoretical contributions.** We begin by highlighting the power of attention, which makes several hard communication problems (EQUALITY, DISJOINTNESS, and INDEX) BAPO-easy. In contrast, we show that a variety of important global problems are BAPO-hard, thus posing a challenge to LLMs with imprecise internal communication. In particular, we prove BAPO-hardness for REACHABILITY, MAJORITY, MATCH3$_n$ [34], UNIQUE, and SETDIFF, with lower bounds on the communication bandwidth required for each problem. On the positive side, we show that chain of thought (CoT) allows us to break down any BAPO-hard problem into a sequence of BAPO-easy steps, suggesting one mechanism for the success of CoT [24, 15, 30]. Specifically, we show that CoT renders constant-bandwidth BAPOs Turing-complete. This enables them to solve any decidable problem given a enough output tokens (although this number might be impractically large).

**Empirical contributions.** Our experimental results confirm the predictive power of the BAPO model: GPT-4o, Claude, and Gemini systematically fail to solve even relatively small instances of BAPO-hard problems, while performing well on BAPO-easy tasks across instance size (as previewed in Figure 1). We also demonstrate how real-world LLM tasks such as aggregating reviews and variable-tracking in code contain BAPO-hard components, namely MAJORITY and REACHABILITY, and thus present a challenge for LLMs. This illustrates the significance of our theoretical work in practice. Supporting our CoT result, the reasoning models o3 and Gemini 2.5 Flash perform very well even on BAPO-hard problems, albeit with a very large number of reasoning tokens. Our code is available at `https://github.com/microsoft/bapo`.

**Implications.** Identifying BAPO-hard and BAPO-easy sub-tasks enables practitioners to anticipate LLM limitations and proactively employ mitigation techniques like inference-time scaling, hybrid architectures, or tool-calling. A key feature of the BAPO model is that it abstracts away the low-level details of transformers and instead focuses on how much information must flow to solve a problem, yielding a characterization that can be applied more broadly and intuitively. Our work also shows how chain-of-thought reasoning can alleviate the communication needs of problems by breaking a problem down into steps requiring only a small amount of information flow, suggesting low bandwidth requirement as an additional objective in learning from reasoning chains. Ultimately, BAPOs offer an explanatory foundation for observed LLM failures on global reasoning problems and can unlock principled innovations to overcome these limitations.

## 2 A communication model of LLMs

The goal of our model is to represent the information processing flow within transformers while abstracting away lower level details. At a high level, recall that a transformer with causal attention

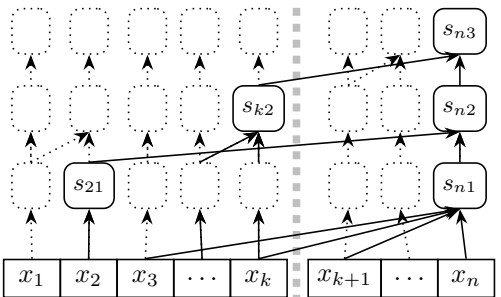

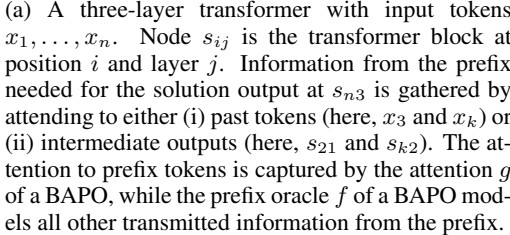

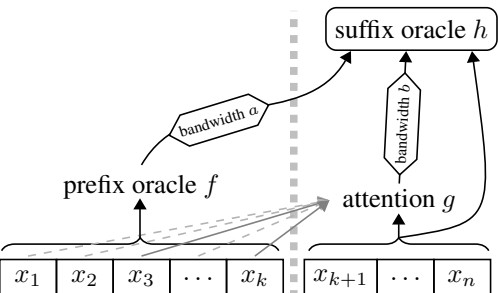

(a) A three-layer transformer with input tokens $x_1, \ldots, x_n$. Node $s_{ij}$ is the transformer block at position $i$ and layer $j$. Information from the prefix needed for the solution output at $s_{n3}$ is gathered by attending to either (i) past tokens (here, $x_3$ and $x_k$) or (ii) intermediate outputs (here, $s_{21}$ and $s_{k2}$). The attention to prefix tokens is captured by the attention $g$ of a BAPO, while the prefix oracle $f$ of a BAPO models all other transmitted information from the prefix.

(b) An $(a, b)$-BAPO computes its result from the output of the prefix oracle $f$ (limited to $a$ bits), attention function $g$ (limited to $b$ tokens), and suffix tokens $k+1, \ldots, n$. The attention function can choose which tokens to attend to as a function of the suffix, but the decision to attend to each prefix token is independent from other prefix tokens. An arbitrary subset $G$ of size $b$ is received by $h$ if $g$ attends to more than $b$ tokens. Every component has access to token indices.

Figure 2: A simplified view of a transformer and our bounded attention prefix oracle (BAPO) model.

makes next-token predictions at every token position in parallel. Additionally, no residual stream knows how far from the end of the input it is or what tokens come later in the input, due to causal attention. As such, if a transformer needs to solve a problem and we split its input into a prefix and suffix, any pre-processing done in the prefix streams should be useful no matter the suffix tokens. Moreover, information about the prefix must be communicated to the suffix streams, as they need to make next-token predictions that may depend on prefix tokens. This communication must occur across *every* possible prefix–suffix split, but for simplicity, we will model and analyze an arbitrary (usually worst case) split.

To make this more concrete, Figure 2a visualizes the computation of the next token $x_{n+1}$ inside a transformer, with nodes contributing to the prediction of $x_{n+1}$ highlighted (other attention weights are 0). A particular prefix–suffix split is shown with a dashed gray line, dividing the input tokens and their residual streams (i.e., the columns above each token). The output stream in the suffix must have all relevant information about the prefix tokens to solve a problem that depends on the whole input. That is, all needed information about the prefix tokens must cross the dashed gray line. This information can either come from (i) attending to input tokens directly (arrows crossing the split into $s_{n1}$) or (ii) attending to intermediate outputs from the prefix streams (arrows crossing the split into $s_{n2}$ and $s_{n3}$). As we have emphasized, any intermediate outputs from prefix layers must be usable for all possible suffixes, since prefix streams cannot depend on future tokens due to causal attention.

Our central hypothesis is that LLMs have a limited ability to exactly communicate a large number of tokens or a large intermediate result across residual streams, thus causing failures on problems that require high information flow. We call this the *effective bandwidth* of an LLM. This hypothesis is informed by prior work that has derived capacity limits on attention heads [12], shown that problems requiring reasoning over many tokens pose a challenge [1], and proved that individual token impacts on attention tend to zero as the input length grows [18].

**BAPOs.** Our model isolates the issue of limited effective bandwidth as it plays a role in LLMs with causal attention. For simplicity, we consider problems with single-token solutions, which includes all decision problems. More formally, we consider tasks where an LLM is prompted with a fixed problem description $\mathcal{P} \in \Sigma^*$ concatenated with an input $x_1 \ldots x_n \in \Sigma^*$, where $\Sigma$ is the token vocabulary. The goal is to produce a solution $y \in \Sigma$, which we represent with the function $p : \Sigma^* \to \Sigma$ with $p(x_1 \ldots x_n) = y$. We begin with an intuitive overview of the model.

**Informal Definition 1.** A *bounded attention prefix oracle* (BAPO; see Figure 2b) must solve a problem given an input split arbitrarily into a prefix and a suffix. A BAPO computes the solution given the suffix, $a$ bits output by a *prefix oracle* $f$ that accesses only the prefix, and $b$ prefix tokens selected individually by a binary attention function $g$, with full positional information. The prefix

oracle $f$ models the intermediate processing in prefix residual streams, which has no access to the suffix due to causal attention. The limits on the output size of $f$ and on the number of tokens $g$ may attend to are the key bandwidth constraints of the model, capturing limited attention head capacity.

Given this intuition, we provide the formal definition of BAPOs (using $\mathbb{N} = \mathbb{Z}_{>0}$), which makes explicit how BAPOs account for positional encodings and what happens if $g$ tries to attend to too many tokens (intuitively, the BAPO must work given any set of $b$ tokens to which $g$ attends).

**Definition 1.** An $(a, b)$-BAPO (*bounded attention prefix oracle*) is defined by a *prefix oracle* $f$ : $\Sigma^* \to \{0,1\}^a$, an *attention function* $g : \Sigma^* \times \mathbb{N} \times \Sigma \times \mathbb{N} \to \{0,1\}$, and a *suffix oracle* $h$ : $\{0,1\}^a \times \cup_{i=0}^b (\Sigma \times \mathbb{N})^i \times \Sigma^* \times \mathbb{N} \to \Sigma$. An $(a, b)$-BAPO *solves* a computational problem $p : \Sigma^* \to \Sigma$ if $h(f(x_1 \ldots x_k), G, x_{k+1} \ldots x_n, k) = p(x_1 \ldots x_n)$ for all $k < n$ and all $G \subseteq \mathbb{G} = \{(x_i, i) : 1 \le i \le k, \ g(x_{k+1} \ldots x_n, k, x_i, i) = 1\}$ with $|G| = \min\{b, |\mathbb{G}|\}$.

We call $a$ the *prefix bandwidth* (measured in bits) and $b$ the *attention bandwidth* (measured in tokens) of the BAPO. We call a problem *BAPO-easy* if it can be solved by a BAPO with constant bandwidths (w.r.t. $n$) and *BAPO-hard* otherwise. If a problem requires bandwidths that scale with $|\Sigma|$, we say it is *BAPO-$\Sigma$-hard*. Note that any problem with $n$-token inputs can be solved by a $(n\lceil\log_2 |\Sigma|\rceil, 0)$-BAPO or by a $(0, n)$-BAPO as the prefix oracle can forward the entire prefix or the attention function can attend to the entire prefix; we call this the trivial upper bound on BAPO complexity. Lastly, given a language $L \subseteq \Sigma^*$, we say that a BAPO *recognizes* $L$ if it solves $p(x) = 1[x \in L]$.

**Assumptions.** We briefly discuss some trade-offs in the underlying assumptions for BAPOs. On the generous side, we assume that the prefix streams and suffix streams have unbounded computational power.[2] However, this fact is tempered by the fact that a BAPO must work for all possible prefix-suffix splits. Regarding the attention function, our model is pairwise in the sense that $g$ can only look at a single prefix token $x_i$ at a time, as in real transformers. BAPOs differ though in that $g$ can base its attention decisions on the entire suffix. However, transformers can compensate for this by communicating information between the suffix streams across multiple layers. Our model also assumes perfect positional encoding, whereas this is a point of failure in real transformers [11]. Finally, the attention $g$ can only operate on the token layer, whereas in transformers, attention can also attend to outputs of subsequent layers. However, this is offset by the ability of the suffix oracle in our model to perform arbitrary computation on the attended tokens. Appendix C explores some variations of the modeling assumptions and how they affect the expressive power of the model.

## 3 BAPO theory

We prove that a variety of important global reasoning problems like graph reachability are BAPO-hard and therefore pose a challenge for LLMs under our effective bandwidth hypothesis. Table 1 summarizes our hardness and tractability results. Strikingly, we also prove that chain of thought enables constant-bandwidth BAPOs to solve all decidable problems, suggesting a reduction in required bandwidth as another mechanism for the empirical success of chain of thought.

### 3.1 BAPO-easy problems

Before turning to problems that BAPOs fail to solve, we first show the kinds of problems that are BAPO-easy, requiring only constant bandwidth. As we will see, these all share the key property that they can be computed from a small summary or local portion of the input.

**One attention token is all you need to solve hard communication problems.** Our first BAPO-easy problems establish a separation from the standard one-way communication model [31] upon which BAPOs are based. BAPOs are at least as powerful: any problem solvable with $a(n)$ bits of one-way communication on $n$-bit inputs is also solvable by a $(a(n), 0)$-BAPO by having the prefix oracle implement the communication protocol. However, adding even just a little attention makes BAPOs strictly more powerful than pure one-way communication. Strikingly, even $(1, 1)$-BAPOs can solve DISJOINTNESS, EQUALITY, and INDEX, which are maximally hard problems for one-way communication requiring $n$ bits of communication [31] (see Appendix A for full definitions).

---

[2]Requiring $f$, $g$, and $h$ to be computable rather than arbitrary has no effect on our results.

Table 1: Overview of our BAPO upper and lower bounds in terms of (prefix bandwidth, attention bandwidth). $n$: input length, $m$: number of edges, $c$: any integer $\geq 3$, $\epsilon$: arbitrary constant in $(0, 1)$, $b(n)$: any $o(n)$ function, $|Q|$: state complexity of the language. Trivial upper bounds: $(n\lceil \log_2 |\Sigma| \rceil, 0)$ or $(0, n)$. Adding chain of thought (CoT) brings the upper bound down to $(2, 3)$ for all decidable problems, but may require a large number of CoT steps.

|  | Problem | Lower bound | Upper bound |
|---|---|---|---|
| BAPO-easy | INDEX (Thm. 1) |  | $(0, 1)$ |
|  | EQUALITY (Thm. 1) |  | $(1, 1)$ |
|  | DISJOINTNESS (Thm. 1) |  | $(1, 1)$ |
|  | MATCH2$_n$ (Thm. 5) |  | $(0, 1)$ |
|  | regular languages (Thm. 2) |  | $(\lceil \log_2 |Q| \rceil, 0)$ |
| BAPO-hard | REACHABILITY (Thm. 3) | $(o(m^{1/c}\log m), o(m^{1-2/c}))$ | trivial |
|  | MAJORITY (Thm. 4) | $(o(\log n), o(n^{1-\epsilon}))$ | $(\lceil \log_2 n \rceil, 0)$ |
|  | MATCH3$_n$ (Thm. 5) | $(o(n/b(n)), b(n))$ | trivial |
| BAPO-$\Sigma$-hard | UNIQUE (Thm. 6) | $(o(|\Sigma|/b(|\Sigma|)), b(|\Sigma|))$ | $(2|\Sigma|, 0)$ |
|  | SETDIFF (Thm. 7) | $(o(|\Sigma|/b(|\Sigma|)), b(|\Sigma|))$ | $(|\Sigma|, 0)$ |

Low BAPO complexity suggests that LLMs should be able to solve these problems well, which is corroborated by our empirical results in Section 4. These problems are also known to be efficiently expressible by transformers [6]. Note also that INDEX is conceptually very similar to the common needle-in-a-haystack benchmark tasks.

**Theorem 1.** DISJOINTNESS *and* EQUALITY *have* $(1, 1)$-*BAPOs and* INDEX *has a* $(0, 1)$-*BAPO.*

*Proof sketch.* Access to token indices allows the attention function to attend to bits that would be counterexamples to EQUALITY or DISJOINTNESS, while the bit output by the prefix oracle is needed when the prefix contains all of $x$ and some of $y$. INDEX is trivial thanks to the attention function, which can pick out the indexed token. See Appendix B for the full proof. □

**Regular languages.** We also show that recognizing any regular language is BAPO-easy, since any string's prefix can be summarized in constant space by a state in the minimal automaton for the language by the Myhill–Nerode theorem [25]. However, this BAPO construction requires prefix bandwidth scaling with the language's *state complexity*, the minimal number of states in an automaton for the language. So, for any LLM with constant effective bandwidth, there may be infinitely many regular languages it cannot recognize. We thus think of the class of regular languages as "fixed-parameter tractable" for BAPOs.

**Theorem 2.** *For any regular $L$ with state complexity $|Q|$, some $(\lceil \log_2 |Q| \rceil, 0)$-BAPO recognizes $L$.*

*Proof.* Let $(Q, \Sigma, \delta, q_0, F)$ be a minimal deterministic finite automaton (DFA) recognizing $L$. The BAPO's prefix oracle sends a binary encoding of the DFA's state after it runs on the prefix. The suffix oracle finishes running the DFA on the suffix and outputs 1 iff the final DFA state is in $F$. □

### 3.2 BAPO-hard problems

We now show that the following problems are BAPO-hard (see Appendix A for formal definitions):

- REACHABILITY: given a directed graph $G$ and two nodes $s, t$, check if $G$ has an $s$–$t$ path.

- MAJORITY: determine whether a bit-string has strictly more ones than zeros.

- MATCH3$_n$: given input $x \in \mathbb{Z}_m^n$, determine whether there are some $i, j \in [n]$ such that $x_n + x_i + x_j \equiv 0 \pmod{m}$. Additionally, MATCH2$_n$ (which we show is BAPO-easy) is the problem of determining whether there is some $i \in [n]$ such that $x_n + x_i \equiv 0 \pmod{m}$. These are last-token versions of MATCH2 and MATCH3 [34].

- UNIQUE (hard w.r.t. $|\Sigma|$): output any token that appears exactly once in the input.

- SETDIFF (hard w.r.t $|\Sigma|$): output a token that is in the first input string but not the second.

We begin our hardness results with the important REACHABILITY problem, which encompasses many natural reasoning tasks, e.g., checking whether a conclusion follows from a chain of implications. We show that limited-bandwidth BAPOs cannot solve REACHABILITY, and our lower bound smoothly trades off between the prefix and attention bandwidth requirements. We provide a high-level sketch of the proof strategy, which is common to all of our hardness proofs. All full proofs omitted from the paper can be found in Appendix B.

**Theorem 3.** *No $(o(m^{1/c} \log m), o(m^{1-2/c}))$-BAPO can solve* REACHABILITY *in graphs with $n$ nodes and $m = \Omega(n)$ edges for any integer constant $c \geq 3$.*

*Proof sketch.* Suppose for a contradiction that some limited-bandwidth BAPO $(f, g, h)$ solves the problem. We construct a family of prefixes and suffixes such that: (1) the prefixes have substantial overlap with each other, (2) the attention function $g$ pays attention to tokens that are common to all prefixes and thus cannot distinguish between them, (3) the number of prefixes is sufficiently large that the prefix oracle $f$ is not one-to-one by the pigeonhole principle, and (4) given any two prefixes for which $f$ collides, we can find a suffix that results in different solutions when concatenated to the colliding prefixes. This construction gives us a contradiction: two instances of the problem with opposite answers which the BAPO cannot distinguish between, as it sees the same suffix, the same output of $f$, and the same set of attended tokens (given some adversarial choice of $G$). □

We suspect this bound is not tight; closing the gap between Theorem 3 and the trivial $(m\lceil \log_2 m \rceil, 0)$- and $(0, m)$-BAPOs is an interesting open problem.

Next, we consider a simple problem where our lower bound is tight, MAJORITY. According to the circuit-based analysis of transformer expressivity that shows they are in $\mathsf{TC}^0$ [23], this problem should be trivial, as it is solved by a single majority gate. However, as we will see in our experiments, LLMs struggle with MAJORITY; here, we show that BAPOs require super-constant bandwidth to solve it and that even near-linear attention bandwidth is insufficient to improve the prefix bandwidth requirement. Our proof follows the same high-level strategy as with REACHABILITY.

**Theorem 4.** *No $(o(\log n), o(n^{1-\epsilon}))$-BAPO can solve* MAJORITY *on length $n$ inputs for any $0 < \epsilon < 1$. This prefix bandwidth is tight: there is a $(\lceil \log_2 n \rceil, 0)$-BAPO solving* MAJORITY.

In Appendix B, we show that increasing the attention bandwidth even beyond near-linear eventually reduces the prefix bandwidth lower bound for MAJORITY.

Turning to the next problem, Sanford et al. [34] showed that 1-layer transformers can efficiently solve MATCH2, but must scale polynomially to solve MATCH3. To keep the single-token output of our problems, we consider the contained subproblems of outputting the last item, which we call MATCH2$_n$ and MATCH3$_n$. We show that limited-bandwidth BAPOs can solve MATCH2$_n$ but not MATCH3$_n$, paralleling the results of Sanford et al. [34]. Our proof again follows the same strategy as before, and as with REACHABILITY, our lower bound trades off between the two bandwidths.

**Theorem 5.** *For any $b(n) = o(n)$ with $b(n) \geq 1$, no $(o(n/b(n)), b(n))$-BAPO can solve* MATCH3$_n$ *over $\mathbb{Z}_{n^2}^n$. In contrast, there is a $(0, 1)$-BAPO for* MATCH2$_n$.

In Appendix C, we show that our BAPO-hardness proofs above are robust to more powerful generalizations of the BAPO model where we allow multiple layers of attention functions operating sequentially, as well as real-valued attention scores rather than binary. While not affecting the hardness of REACHABILITY, MAJORITY, or MATCH3$_n$, this augmented model is able to solve the (multi-hop) induction heads task [26], which otherwise appears to be BAPO-$\Sigma$-hard.

Finally, we show two problems are BAPO-$\Sigma$-hard, starting with UNIQUE, the problem of finding an item that appears exactly once in a sequence. Here, the difficulty of the problem is parametrized by the size of the token vocabulary $\Sigma$ rather than the length of the input $n$. A very similar approach applies to SETDIFF, for which we find the same bound.

**Theorem 6.** *Let $k = |\Sigma|$. For any $b(k) = o(k)$ with $b(k) \geq 1$, no $(o(k/b(k)), b(k))$-BAPO can solve* UNIQUE. *This is tight for $b(k) = O(1)$, as there is a $(2k, 0)$-BAPO solving* UNIQUE.

**Theorem 7.** *Let $k = |\Sigma|$. For any $b(k) = o(k)$ with $b(k) \geq 1$, no $(o(k/b(k)), b(k))$-BAPO can solve* SETDIFF. *This is tight for $b(k) = O(1)$, as there is a $(k, 0)$-BAPO solving* SETDIFF.

### 3.3 BAPOs with chain of thought

We now show that any (decidable) BAPO-hard problem can be broken down into a sequence of BAPO-easy steps in the spirit of *chain of thought* (CoT) [43]. Previous work has shown how CoT makes transformers Turing-complete [24]. We show that CoT has an even stronger benefit: adding CoT to BAPOs lets them solve all decidable problems *with constant bandwidth*, indicating that even LLMs with constant effective bandwidth are Turing-complete with CoT.

To formalize CoT in the BAPO setting, we repeatedly apply a fixed BAPO to the input concatenated with the BAPO's previous output tokens, just as in auto-regressive decoding in LLMs.

**Definition 2.** An $(a, b)$-*bounded attention prefix oracle with chain of thought* (BAPO-CoT) that solves a computational problem $p : \Sigma^* \to \Sigma$ is an $(a, b)$-BAPO over the token set $\Gamma \supseteq \Sigma \cup \{\Box\}$ that solves some computational problem $p' : \Gamma^* \to \Gamma$ such that for all inputs $x \in \Sigma^*$ to $p$, there exists some sequence of strings $s_1, \ldots, s_m \in \Gamma^*$ with the following properties. (1) $s_1 = x$ (the BAPO-CoT starts with $x$ as its input), (2) $s_{i+1} = s_i p'(s_i)$ for all $i = 1, \ldots, m - 1$ (at each step, it produces some chain-of-thought token), (3) $p'(s_i) = \Box$ if and only if $i = m - 1$ (at the last step only, it outputs the halt token), and (4) $p'(s_{m-2}) = p(x)$ (it solves the problem before halting).

Let $L^{\leq n} = \{x \in L : |x| \leq n\}$. We show that low-bandwidth BAPO-CoTs exist for any decidable problem; the core idea is that simulating a single Turing machine step is a low-bandwidth problem.

**Theorem 8.** *Let $L$ be a language decided by a Turing machine with $s(n)$ space and input alphabet $\Sigma = \{0, 1\}$. For any $n$, there exists a $(2, 3)$-BAPO-CoT that recognizes $L^{\leq n}$.*

*Proof sketch.* The BAPO-CoT simulates a Turing machine $M$ by writing out the contents of the tape and the state at every step of $M$'s execution. The attention function attends to the prefix to retrieve the current state and the prefix oracle passes along the bit under the tape head (if they are not in the suffix), allowing the suffix oracle to simulate a step of $M$. See Appendix B for details. ∎

## 4 BAPO complexity predicts empirical LLM failures

Our experiments test whether BAPO complexity predicts LLM failures. Supporting our effective bandwidth hypothesis, we find that LLMs across model families consistently struggle with BAPO-hard problems and usually succeed on BAPO-easy problems. We choose three big model families (GPT [27], Gemini [16] and Claude [3]) and focus first on model versions without (latent) reasoning chains to align with the single-token-output BAPO model.

Except for INDEX and the BAPO-$\Sigma$-hard problems UNIQUE and SETDIFF, all problems have yes/no answers (so guessing achieves 50% accuracy). We designed problem instances so that there would be no obvious shortcut or heuristic, pushing models to fully consider each problem. LLMs are fed with inputs of various lengths $n$, where $n$ corresponds to the parameter specified in each problem's definition. For all problems, we generate 100 i.i.d. instances and report average accuracy along with the 95% $t$-test confidence interval. All data generating distributions, prompts, and model details are available in Appendix D.1 and our code is available at `https://github.com/microsoft/bapo`.

### 4.1 BAPO hardness aligns well with LLM failures

Figure 3 shows the accuracy of LLMs across six different tasks (see Appendix D.2 for additional problems). The top row shows three BAPO-easy problems and the bottom row three BAPO-hard problems (cf. Table 1). Performance is low or rapidly dropping for BAPO-hard problems; in particular, there is no LLM that performs well for all $n$. In contrast, most LLMs perform consistently well across all $n$ on BAPO-easy problems, with MATCH2 appearing the hardest. We suspect that representational issues interfere in this setting, as the LLM needs perfect understanding of integers.

Comparing models of different scales (solid vs. dashed lines), we can see that in line with the typical observations, larger models appear to perform better overall. However, even with increased scale, no model is able to avoid the degradation our BAPO framework predicts.

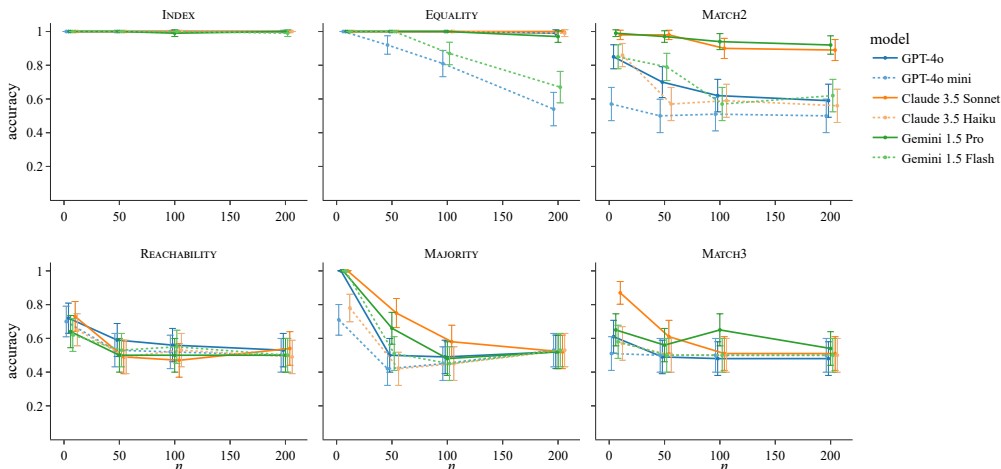

Figure 3: BAPO-hard problems (bottom row) show much larger drops in accuracy compared to BAPO-easy problems (top row). Not even large LLMs can solve BAPO-hard problems at length 200 with an accuracy above random guessing.

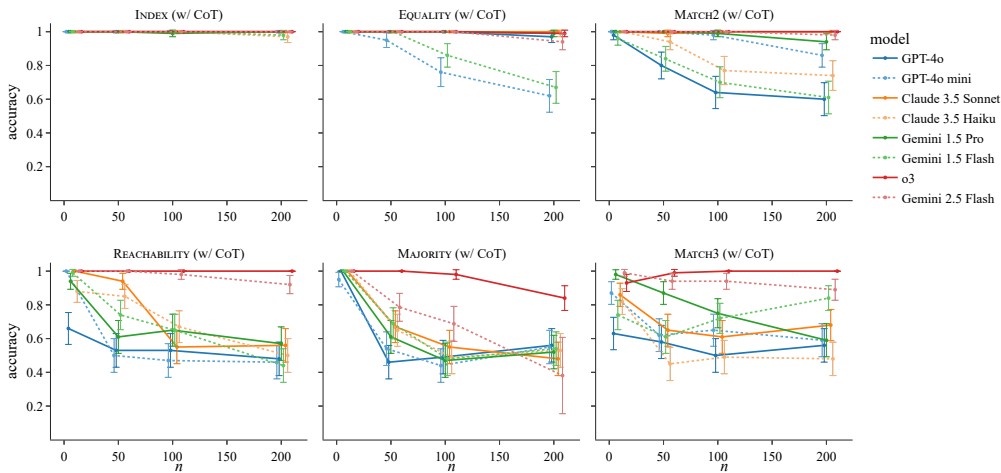

Figure 4: Adding CoT can help LLMs do better on BAPO-hard problems, but substantial performance drops still occur with limited CoT budget (soft limit of 250 words for non-reasoning models). Without imposing a limit on their internal reasoning, o3 and, to a lesser extent, Gemini 2.5 Flash perform extremely well (see Appendix D.1.2 for their CoT token counts, often in the thousands).

## 4.2 Chain-of-thought reasoning helps on BAPO-hard problems

Given how powerful BAPO-CoTs are in theory, the obvious question is whether CoT improves performance on BAPO-hard tasks. To test this, we prompted each LLM to perform CoT before producing the answer, with a soft limit of 250 words. We also tested o3 [28] and Gemini 2.5 Flash [17], which use (potentially many) internal chain of thought tokens. As Figure 4 shows, CoT modestly improves non-reasoning LLM performance on BAPO-hard problems for smaller input sizes $n$. The fact that issues still persist indicates that these LLMs may not be applying good low-bandwidth CoT procedures, or that they may require more reasoning tokens. Indeed, without the limit on CoT tokens, o3 and Gemini 2.5 Flash succeed on BAPO-hard problems. This is likely due to the much larger number of CoT tokens they use (over 10k in some cases; see Appendix D.1.2) and the fact that they are (presumably) trained to use CoT tokens effectively.

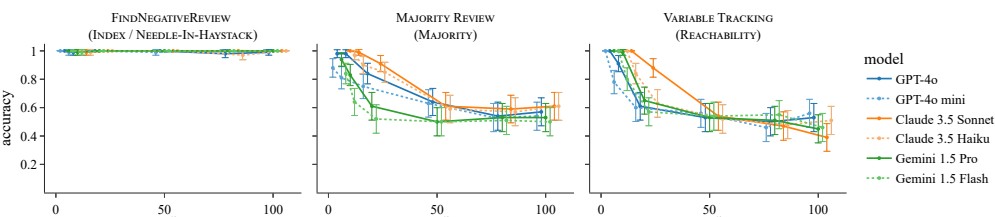

Figure 5: There is good evidence that BAPO-difficulty translates to real-world settings. LLMs can solve real-world tasks that contain BAPO-easy problems (left plot) with much greater accuracy than BAPO-hard problems (two plots on the right).

### 4.3 BAPO hardness in real-world tasks

We finally turn to a set of experiments that examine real-world tasks corresponding to BAPO problems. For the first domain, inspired by the ZeroScrolls benchmark [38], we consider hotel reviews from the SPACE dataset [2] and either ask the LLM to find a negative review in a collection of positive reviews (analogous to the BAPO-easy INDEX problem) or decide whether the majority of reviews are positive (analogous to the BAPO-hard MAJORITY problem). We ensure that a baseline LLM can determine the sentiment of each review. The second domain is programming, where we define two chains of assignments and ask whether the final variable has value "a" or "b", extending the variable tracking task from the RULER benchmark [20]. This task is a special case of REACHA-BILITY, one of our BAPO-hard problems. See Appendix D.1.3 for experiment and data details. The plots in Figure 5 show that again, BAPO-hardness is a good predictor of LLM performance.

## 5 Related work

Theoretically, limitations of transformers have been studied via communication complexity [35, 29], circuit analysis [23, 39] and parallel computation frameworks [35, 37], among other methods. We contribute to this toolbox with the BAPO model, which provides a natural way to study how causal attention exacerbates limits on information flow in LLM architectures.

Among the best known results on transformer expressivity is that transformers are in log-space uniform $TC^0$ [23, 39]. Strobl et al. [40] survey known theoretical results showing an inherent expressivity gap for transformers to recognize certain formal languages. For instance, Hahn [18] show that transformers with one-hot hard attention cannot solve PARITY and 2DYCK, while Bhattamishra et al. [6] show that a two-layer transformer can solve INDEX, EQUALITY, and DISJOINTNESS. RASP [44] offers a higher-level way to establish similar upper bounds by constructing a transformer as a program, although without causal masking. Some expressivity analyses depend on the size of the transformer: Sanford et al. [35], Fagnou et al. [14] derive logarithmic lower bounds on the number of transformer layers required for graph and entity tracking tasks (related to similar bounds for map-reduce [32]), while Sanford et al. [34] show that a single transformer layer can efficiently reason over pairs of tokens (MATCH2) but not triplets (MATCH3). However, transformer expressivity does not always align with empirical observations of LLM performance. Rather than characterizing theoretical expressivity, the BAPO model captures high-level information flow, which we hypothesize underlies many problem-solving failures in practice.

Even when a task is, in principle, solvable by a transformer, it might still be hard to learn [8]. Edelman et al. [12] argue that attention in transformers tends to represent dependencies among only a small number of tokens, causing failures for global problems. Hahn and Rofin [19] show that although PARITY is representable, it is hard to learn a length-generalizing solution because the training loss landscape is highly sensitive to all the inputs. Similarly, Thomm et al. [42] show that LLMs are data inefficient over compositional problems and Liu et al. [22] argue that transformers learn short-cuts to simulating finite-state automata. Our BAPO model captures this representability-learnability gap by positing that although architecturally the communication bandwidth from prefix tokens of a transformer can be large, the effective communication bandwidth in LLMs is very limited.

Lastly, the success of CoT [43] has also been analyzed theoretically. Merrill and Sabharwal [24] showed that CoT makes transformers Turing-complete, and Feng et al. [15] argue that CoT enables

transformers to solve dynamic programming problems that bounded-depth transformers cannot. Our theory shows that CoT has another benefit, as it dramatically lowers bandwidth requirements: any Turing machine can be simulated by a constant-bandwidth BAPO-CoT.

## 6  Discussion

Our finding that the effective bandwidths of LLMs are small despite their massive size suggests that simply adding more layers, attention heads, or embedding dimensions might not translate directly to higher BAPO bandwidth. We still lack a full understanding of what causes this severely limited bandwidth. It might even be a feature rather than a bug: low effective bandwidth may aid flexible and generalizable next-token prediction, and there could be a tradeoff between generalization ability (required for natural language) and exact input representation (required for global reasoning).

Beyond the precise mathematical framework of BAPOs, applying the intuition that lower-bandwidth problems are easier for LLMs can help us understand their successes. For instance, many in-context learning tasks can be solved by a $k$-nearest neighbor approach, matching a new instance to a small number of in-context examples. This is a procedure whose bandwidth requirement does not scale with the number of in-context examples, but whose accuracy does; this provides a possible explanation for the success of LLMs on such tasks. As another example, needle-in-a-haystack tasks commonly used to benchmark LLMs also require a small amount of cross-stream communication.

**Lowering bandwidth.**  Our model also deepens the understanding of CoT by proving that it reduces the communication requirements of problems—although the number of reasoning steps can be impractically large. This motivates future work to take better advantage of the bandwidth-lowering benefits of CoT or directly optimizing for low bandwidth as part of the training objective when fine-tuning on reasoning chains. Our work also motivates the investigation of methods beyond inference time scaling for reducing the communication burden of problems. For some problems, pre-processing such as simplifying inputs [45] or retrieval [21] may reduce communication load.

**Refining BAPO.**  Another future direction is in exploring variations of the BAPO model to help it align more closely with the behavior of LLMs. For example, the induction heads task [26, 37] is thought to be an important mechanism for transformers, but appears challenging for BAPOs. However, we show in Appendix C that a BAPO with multiple layers of score-based attention can perform induction heads, while preserving the BAPO-hardness of REACHABILITY, MAJORITY, and MATCH3$_n$. This suggests these problems are fundamentally hard, and that others lie between the basic BAPO model and LLM capabilities. Future work can explore how these and other BAPO variants affect BAPO-hardness and which problems are fundamentally high-bandwidth.

**Limitations.**  See Section 2 for discussion of ways in which the BAPO model does not faithfully represent transformer computation. It also does not capture all failure modes of LLMs, such as tokenization-driven errors; thus, BAPO-easiness is not a guarantee that LLMs can solve a task. We also do not know the root cause of effective bandwidth limits on LLMs. Finally, many of our lower bounds are loose, and there are many problems whose BAPO bandwidths have yet to be explored.

## 7  Conclusions

We introduced the BAPO model of bandwidth-limited computation, designed to quantify and analyze hypothesized limits on the cross-stream communication of transformer-based LLMs. On the theoretical side, we categorize a variety of problems as BAPO-easy, requiring only constant bandwidth, and BAPO-hard, requiring super-constant bandwidth. This dividing line aligns well with problems that modern trillion-parameter-scale LLMs consistently struggle with, supporting the hypothesis that they are constrained in their internal communication and indicating that their effective bandwidth for problem-solving is a small constant. For practitioners, the BAPO framework offers a new lens through which they can view their LLM tasks, possibly opting for mitigation strategies such as tool calling and reasoning in cases where they suspect failures due to BAPO-hardness. Understanding that limited communication bandwidth is at the heart of why LLMs fail to reason globally also unlocks a new set of directions for future work, such as different architectures, reasoning algorithms, or training paradigms.

## Acknowledgments

We are grateful to Doug Burger, Philippe Laban, Suriya Gunasekar, Daniel Hsu, Besmira Nushi, Clayton Sanford, Siddharth Suri, Dawen Liang, Harald Steck, Chinmaya Kausik, Nathan Kallus, the MSR AI Interaction and Learning group, and the Netflix Machine Learning Inference Research group for helpful discussions and feedback.

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

# A  Full problem definitions

**Definition 3.** DISJOINTNESS: $\{0,1\}^n \times \{0,1\}^n \to \{0,1\}$ is the problem of finding if two sets represented as bit-strings are disjoint, with $\text{DISJOINTNESS}(x,y) = \bigwedge_{i \in [n]} \neg(x_i \wedge y_i)$. We encode a DISJOINTNESS instance $(x,y)$ with the string $x|y$ over $\Sigma = \{0,1,|\}$.

**Definition 4.** EQUALITY: $\{0,1\}^n \times \{0,1\}^n \to \{0,1\}$ is the problem of finding if two bit-strings are equal, with $\text{EQUALITY}(x,y) = 1[x = y]$. We encode an EQUALITY instance $(x,y)$ with the string $x|y$ over $\Sigma = \{0,1,|\}$.

**Definition 5.** INDEX: $\{0,1\}^n \times [n] \to \{0,1\}$ is the problem of identifying the bit at a given index into the input, with $\text{INDEX}(x,i) = x_i$. We encode an INDEX instance $(x,i)$ with the string $xi$ over $\Sigma = \{0,1\} \cup [n]$.

**Definition 6.** REACHABILITY: $([n] \times [n])^m \times [n] \times [n] \to \{0,1\}$ is the problem of determining if there is path from $s$ to $t$ in a directed graph $G$ with $n$ nodes and $m$ edges. To encode the problem with perfect tokenization, let $\Sigma = [n] \times [n] \cup [n]$ where the token $(i,j) \in \Sigma$ represents edge $(i,j)$ and the integer tokens represent nodes. An instance of REACHABILITY is specified by the edge list of $G$ in arbitrary order followed by the nodes $s$ and $t$.

**Definition 7.** MAJORITY: $\{0,1\}^n \to \{0,1\}$ is the problem of determining whether the input has strictly more ones than zeros, with $\text{MAJORITY}(x) = 1\left[\sum_{i \in [n]} x_i > n/2\right]$.

**Definition 8.** Given input $x \in \mathbb{Z}_m^n$, $\text{MATCH3}_n$ is the problem of determining whether there are some $i, j \in [n]$ such that $x_n + x_i + x_j \equiv 0 \pmod{m}$. Additionally, $\text{MATCH2}_n$ is the problem of determining whether there is some $i \in [n]$ such that $x_n + x_i \equiv 0 \pmod{m}$. We encode instances of these problems with the string $x$ over $\Sigma = \mathbb{Z}_m$.

**Definition 9.** UNIQUE: $\Sigma^* \to \Sigma \cup \{\emptyset\}$ is the problem of identifying any unique token in the input. That is, $\text{UNIQUE}(x_1 \ldots x_n) = x_i$ s.t. $\sum_{j=1}^{n} 1[x_j = x_i] = 1$, or $\emptyset$ if no such $x_i$ exists.

**Definition 10.** SETDIFF: $\Sigma^* \times \Sigma^* \to \Sigma \cup \{\emptyset\}$ is the problem of identifying any token in the first part of the input that does not appear in the second part. That is, $\text{SETDIFF}(c_1 \ldots c_n, d_1 \ldots d_m) = c_i$ for some $i \in [n]$ such that $c_i \neq d_j$ for all $j \in [m]$, or $\emptyset$ if no such element exists. We encode a SETDIFF instance $(c,d) \in \Sigma^* \times \Sigma^*$ with the string $c|d$ over $\Sigma \cup \{|\}$.

# B  Proofs

*Proof of Theorem 1.* For INDEX instances given as $s = xi$, the idea is simple: we set $g(s_{k+1} \ldots s_n, k, s_j, j) = 1$ if and only if $j = s_n$. The suffix oracle returns whatever token is attended to—unless the index is in the suffix (i.e., $s_n \geq k+1$), in which case the suffix oracle immediately returns the bit at position $s_n$, which lies in the suffix and is therefore visible without using attention. No output from $f$ is required.

For a DISJOINTNESS instance encoded as $s = x|y$, the key is that we only need a single counterexample bit where $x_i = y_i = 1$ to conclude that $x$ and $y$ are not disjoint. If no such counterexample exists, then they are disjoint. We can construct a $(1,1)$-BAPO using this fact. The prefix oracle outputs 0 ("no counterexample found") if (a) the token $|$ is not in the prefix or (b) if the token $|$ is in the prefix and the starting bits of $y$ present in the prefix are disjoint with the corresponding bits of $x$. If there is a counterexample bit in $y$ visible in the prefix, the prefix oracle outputs 1. If the suffix oracle receives a 1 from $f$, it always outputs 0 indicating $x$ and $y$ are not disjoint. Meanwhile, the attention function attends to any prefix bits that would be a counterexample to the disjointness of $x$ and $y$; i.e., $g(s_{k+1} \ldots s_n, k, s_i, i) = 1$ if and only if either (a) $|$ is not in the suffix and $i' = i + (k + |s_{k+1} \ldots s_n| - 1)/2 + 1 \geq k+1$ (i.e., the token we need to compare to $s_i$ is in the suffix) and $s_{i'} = s_i = 1$ or (b) $|$ is in the suffix and $s_{i'} = s_i = 1$. If the attended token set $G$ is nonempty, then $x$ and $y$ are not disjoint and the suffix oracle outputs 0. If the attended token set is empty and $|$ is in the suffix, the suffix oracle can check the portion of $x$ present in the suffix against the corresponding bits of $y$. If no counterexample bits are found (by $f$, $g$, or $h$), then none exists and the suffix oracle outputs 1.

For EQUALITY, the same idea applies: we only need one counterexample $x_i \neq y_i$ to conclude that $x \neq y$. The construction is precisely the same as for DISJOINTNESS, except we look for counterexamples where $x_i \neq y_i$ rather than $x_i = y_i = 1$. $\qquad\square$

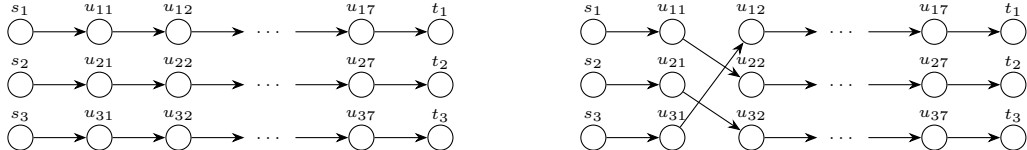

Figure 6: Left: the graph $P$ from the proof of Theorem 3 for $p = c = 3$, so $n = p^c = 27$ and $m = p^c - p = 24$. Right: $P$ with the permutation $\pi = 231$ applied to the targets of the second-layer edges. Applying any non-identity permutation to a single layer changes the connectivity of at least one $s$–$t$ pair.

*Proof of Theorem 3.* Suppose such a BAPO exists for a contradiction, and let $f$ and $g$ be its prefix oracle and attention function, with bandwidths $a = o(m^{1/c} \log m)$ and $b = o(m^{1-2/c})$, respectively. We will construct a family of prefixes and suffixes where the prefixes contain the entire graph and the suffixes contain the nodes $s$ and $t$, taking care that the graphs in different prefixes are mostly identical to saturate attention.

Let $n = p^c$ for some integer $p$ for simplicity (if $n$ is not expressible as a $c$th power of an integer, the argument would involve various ceilings or floors, but these do not affect the asymptotics). Let $P$ be the graph consisting of $p$ disjoint directed paths with start nodes $s_1, \ldots, s_p$ and target nodes $t_1, \ldots, t_p$, where each path has exactly $p^{c-1}$ nodes. Then, for each $i \in [p]$ and $j \in [p^{c-1} - 2]$, let $u_{ij}$ be the unique node at distance $j$ from $s_i$. See Figure 6 for a visualization of this graph with $p = c = 3$. We will keep these node labels fixed but modify $P$ by permuting edges based on the attention function $g$ to render attention useless. Note that we can apply a permutation $\pi \in S_p$ to the target nodes of the edges departing from $u_{1j}, \ldots, u_{pj}$ while maintaining the property that $P$ is a disjoint collection of length $p^{c-1}$ paths each starting from an $s$ node and ending at a $t$ node, although the permutation changes which $s$–$t$ pairs are connected (see Figure 6, right). To construct our shared prefixes that saturate the attention function $g$, we will need to take care to place edges at consistent indices in the prefixes. To this end, we fix the edge order to first list the edges departing from $s_1, \ldots, s_p$, then $u_{11}, \ldots, u_{p1}$, then $u_{12}, \ldots, u_{p2}$, etc. Call this the canonical edge order and let $I(u)$ denote the index in the canonical order where the edge departing $u$ is placed. Note that the length of the full edge list is $m = p^c - p$. Given these observations, construct the shared prefix graph $P^*$ as follows:

1. Initialize $P^*$ to have the same set of nodes as $P$. Initialize $S$ to be the set of all $s_i$ and $u_{ij}$ nodes, which will store the current set of nodes that still need an outgoing edge.

2. For each pair $(i, j) \in [p] \times [p]$:

   (a) For $\ell = 1, \ldots, b$:
      i. If there is some node $u$ in $S$ and some node $v$ in the layer to the right of $u$ such that $g(s_i t_j, m, (u, v), I(u)) = 1$ and $v$ has in-degree 0 in $P^*$: add $(u, v)$ to $P^*$ and remove $u$ from $S$.
      *Check every feasible edge we could add; if $g$ wants to attend to any edge $(u, v)$, add it to $P^*$.*
      ii. Else: exit the inner for loop.
      *If there are no remaining edges $g$ would attend to given suffix $s_i t_j$, then we can stop, as attention is now saturated for this suffix.*

3. Let $E^*$ be the set of edges in $P^*$ at this point in the algorithm. These will be shared among all prefixes to saturate attention. To complete the paths in $P^*$ arbitrarily, connect each node with out-degree 0 (that is not a $t$ node) to the first node with in-degree 0 in the next layer.

This procedure produces a graph $P^*$ which can be reached by starting with $P$ and permuting the targets of the edges in each layer by some $\pi$, since each edge we add connects a node with current out-degree 0 to a node in the next layer with current in-degree 0. That is, $P^*$ is also a disjoint collection of $p$ paths of length $p^{a-1}$ each connecting some $s_i$ to some $t_j$. We have placed edges in $P^*$ such that for any suffix $s_i t_j$ with $i, j \in [p]$, there exists some attended token set $G$ where *every*

edge in $G$ is in $E^*$, so we can modify the other edges in $P^*$ without alerting the attention. Each iteration of the outer for loop ensures one suffix has its attention masked; if we complete $b$ iterations of the for loop, then we have ensured there is some $G$ with $|G| = b$ that contains only edges in $E^*$, and if we need to exit early due to the nonexistence of an attended edge in line 2(a)i, then no other edges we end up placing in $P^*$ will ever be attended to given that suffix (as 2(a)i checks *every* feasible edge). Moreover, the total number of edges in $E^*$ is at most $bp^2 = o(m^{1-2/c}p^2) = o((p^c)^{1-2/c}p^2) = o(p^c)$ (since there are $m = p^c - p < p^c$ edges in $P^*$).

Now, since there are $p^{c-1} - 1$ layers of edges in $P^*$, there must be some layer with $\Theta(p)$ edges not contained in $E^*$—otherwise we would be able to find $\Theta(p^c)$ edges in $E^*$, a contradiction. Call this layer $j^*$ and let $d = \Theta(p)$ be the number of edges in layer $j^*$ that are not in $E^*$. Construct a family of graphs $\mathcal{P}$ of size $d!$ by taking $P^*$ and applying each permutation $\pi \in S_d$ to the targets of edges in layer $j^*$ that are not in $E^*$. We now have our full collection of prefixes and suffixes: the $d!$ prefixes are the canonical edge lists of each graph in $\mathcal{P}$ and the $p^2$ suffixes are every $s_i t_j$ pair.

By Stirling's approximation and the fact that $d = \Theta(p)$, $\log_2(d!) = \Theta(d \log d) = \Theta(p \log p)$. Thus, since $f$ has bandwidth $o(m^{1/c} \log m) = o(p \log p)$, $f$ must have a collision on the size $d!$ family of prefixes generated by $\mathcal{P}$ (as it takes $\log_2(d!) = \Theta(p \log p)$ bits to distinguish between the prefixes). These two colliding graphs are generated by different permutations on the edges in layer $j^*$, so there is some pair of start and target nodes $s^*$–$t^*$ connected in one of the graphs but not the other (as the edges in all other layers are identical). Given the suffix $s^*t^*$, we find that the BAPO fails to distinguish between the colliding graphs: it gets the same output from $f$ and there is some attended token set $G$ that contains only edges in $E^*$, which are identical in the two graphs. Given this adversarial $G$ containing only attention-saturating edges, the suffix oracle $h$ gets identical inputs: the same output from $f$, the same $G$, the same suffix $s^*t^*$ and the same prefix size $m$. In this case, the BAPO fails to solve the problem in one of the two graphs, as one has an $s^*$–$t^*$ path and the other does not. □

*Proof of Theorem 4.* First, a $(\lceil \log_2 n \rceil, 0)$-BAPO can solve MAJORITY by having $f(x)$ output the number of 1's in $x$, which takes at most $\lceil \log_2 n \rceil$ bits and requires no attention tokens.

Now suppose for a contradiction that $(f, g, h)$ is a BAPO with prefix bandwidth $a = o(\log n)$ and attention bandwidth $b = o(n^{1-\epsilon})$ that solves MAJORITY on inputs of length $n$. Let $m$ be any positive integer and let $\ell = m^{(1-\epsilon)/\epsilon}$ (for simplicity assume $\ell$ is an integer; otherwise we could take a ceiling without disrupting the proof). The proof follows three high-level steps: (1) set up the structure of prefix–suffix pairs, (2) design the prefixes so that attention cannot distinguish between them, (3) find a colliding pair of prefixes and suffixes that the BAPO cannot distinguish between but which have different MAJORITY answers.

Given any $\pi \in S_m$ (the permutations on $[m]$), define $X_\pi = \{(\pi(0^{\ell m}1^{\ell m}0^k1^{m-k}), 0^{m-k-1}1^{k+1})\}_{k=0}^{m-1}$. Note that every prefix–suffix pair in $X_\pi$ has total length $n = 2(\ell+1)m = 2m^{1+(1-\epsilon)/\epsilon} + 2m$, with $(\ell+1)m+1$ ones and $(\ell+1)m-1$ zeros; so all of them have a majority of ones. Let $s_k$ be the suffix in $X_\pi$ with $k + 1$ ones and let $S = \{s_k\}_{k=0}^{m-1}$. We will show show to pick $\pi$ based on $g$ such that $X_\pi$ fools the BAPO. We will give the BAPO $\ell = m^{(1-\epsilon)/\epsilon}$ attention tokens. Note that $m^{(1-\epsilon)/\epsilon} = \Theta(n^{1-\epsilon})$, as $(m^{1+(1-\epsilon)/\epsilon})^{1-\epsilon} = m^{(1-\epsilon)/\epsilon}$.

For any suffix $s$, define $\Gamma_0(s) = \{i \in [(2\ell+1)m] : g(s, (2\ell+1)m, 0, i) = 1\}$ to be the set of prefix indices where $g$ selects 0's given suffix $s$, and define $\Gamma_1(s)$ similarly to be the set of prefix indices where $g$ selects 1's. We will pick $\pi$ so that for every $s \in S$, $\pi$ permutes the prefixes so that it places leading zeros and ones in at least $\ell$ indices selected by $g$ (or such that $g$ selects only indices masked by leading zeros and ones) *for every suffix* in $X_\pi$. We construct that $\pi$ with the following procedure.

1. Initialize $\Pi_0 = \emptyset$ and $\Pi_1 = \emptyset$.
   *These sets store indices to place leading zeros and ones so that we fool $g$ for all suffixes.*

2. For $k = 0, \ldots, m-1$:

   (a) While $|\Gamma_0(s_k) \cap \Pi_0| + |\Gamma_1(s_k) \cap \Pi_1| < \ell$:
       *If this is false, we have succeeded in masking $\geq \ell$ positions selected by $g$ given $s_k$.*
       i. If $\exists i \in \Gamma_0(s_k) \setminus (\Pi_0 \cup \Pi_1)$: add $i$ to $\Pi_0$.
          *Mask a position where $g$ selects a zero.*

    ii. Else if $\exists i \in \Gamma_1(s_k) \setminus (\Pi_0 \cup \Pi_1)$: add $i$ to $\Pi_1$.
    *Mask a position where $g$ selects a one.*

    iii. Else: exit the while loop.
    *There are no unmasked positions selected by $g$ on $s_k$, so we have succeeded on $s_k$.*

3. Let $i = 1$. For each $j \in \Pi_0$, set $\pi(i) = j$ and increment $i$.
*Make $\pi$ permute leading zeros into the masking indices we have picked.*

4. Let $i = \ell m + 1$. For each $j \in \Pi_1$, set $\pi(i) = j$ and increment $i$.
*Make $\pi$ permute leading ones into the masking indices we have picked.*

5. Fill in $\pi$ where not yet defined with the remaining indices in order.

This procedure terminates, since each iteration of the while loop increases the combined sizes of the intersections in (a) by 1 or exists the while loop. Additionally, it generates a valid permutation and $\Pi_0$ and $\Pi_1$ are disjoint, since we only ever add indices $i$ which are in neither of them to exactly one of them. Next, for every $(\pi(p), s) \in X_\pi$, either: (1) there are at least $\ell$ indices $I$ selected by $g$ on input $(\pi(p), s)$ with $I \subseteq \Pi_0 \cup \Pi_1$ or (2) on input $(\pi(p), s)$, $g$ selects only indices in $\Pi_0 \cup \Pi_1$. Case (1) occurs if we are able to complete the while loop for $s$ without ever hitting the else in iii, since then we will have $\ell$ such indices in $\Pi_0$ and $\Pi_1$. In 3, we ensure the indices where $g$ wishes to select zeros are masked with leading zeros; in 4, we ensure the indices where $g$ wishes to select ones are masked with leading ones. Case (2) occurs if we hit the else in iii, as in that case, there are no prefix indices selected by $g$ which are not already in $\Pi_0 \cup \Pi_1$ and thus all selected indices will be masked by leading zeros and ones.

This $\pi$ gives us a set $X_\pi$ where the set of attended tokens $G$ is entirely useless over *every* prefix-suffix combination, in the worst case over adversarial choices of which $\ell$ tokens are attended to. That is, for any prefix $\pi(p)$ and any suffix $s$ from $X_\pi$ that may be from different pairs, $G_{(\pi(p),s)}$ (the attended token set for the given prefix–suffix combination) can be identical: an adversary picking which $\ell$ tokens are attended to can pick all $\ell$ of them to come from indices masked by leading zeros and ones (i.e., indices from $\Pi_0 \cup \Pi_1$), which are identical across all $m$ prefixes in $X_\pi$, as the prefixes only differ in indices not in $\Pi_0 \cup \Pi_1$.

Now, there are $m$ distinct strings in $X_\pi$. Since the prefix bandwidth of $f$ is $o(\log n) = o(\log m^{1+(1-\epsilon)/\epsilon}) = o(\log m)$, there are fewer than $2^{\log_2 m} = m$ distinct outputs of $f$, so by the pigeonhole principle there are two elements $(a, b), (c, d) \in X_\pi$ where $f(a) = f(c)$. If $(a, b)$ has $k + 1$ ones in $b$ and $(c, d)$ has $k' + 1$ ones in $d$ (without loss of generality assuming $k' > k$), then the string $ab$ has a majority of ones ($\ell m + m - k + k' + 1 > (\ell + 1)m + 1$ ones total) but $cb$ does not ($\ell m + k' + m - k - 1 \geq (\ell + 1)m$ zeros total). Thus, consider the inputs $(a, b)$ and $(c, b)$. The prefix oracle $f$ outputs the same thing in both cases, and as we have seen that $G_{(a,b)}$ can equal $G_{(c,b)}$. Moreover, the suffix oracle gets the same suffix in both cases, so there exist attended token sets where the suffix oracle outputs the same answer on both input pairs (having received all the same inputs: same prefix oracle output, same suffix, same attended tokens, same split index). But $(a, b)$ and $(c, b)$ have opposite answers to MAJORITY: $(a, b)$ has a majority of ones and $(c, b)$ does not. Thus the BAPO answers one of these instances incorrectly for some attended token set. $\qquad \square$

While near-linear attention does not help the required asymptotic prefix bandwidth over zero attention, our proof does result in a weaker lower bound on prefix bandwidth with even more attention.

**Corollary 1.** *No $(o(\log \log n), o(n/\log n))$-BAPO can solve* MAJORITY *on length $n$ inputs.*

*Proof.* Now we use even more masking bits, $m2^m$ zeros and ones, so $n = m2^{m+1} + 2m$. With $2^m = \Theta(n/\log n)$ attention tokens, using the same argument shows that $o(\log \log n) = o(\log m)$ prefix bandwidth can cause a collision. $\qquad \square$

We suspect that this lower bound is loose, and that in fact $\log n$ prefix bandwidth is still required even with $\Theta(n/\log n)$ attention. However, we can increase the attention bandwidth even more (all the way to $\Theta(n)$, but still less than $n$) so that we do lower the asymptotic prefix bandwidth requirement. For instance, with $n - \log n$ attention tokens, we only need $\Omega(\log \log n)$ prefix bandwidth.

**Proposition 1.** *For any $1 \leq a(n) = o(n)$, there is a $(\lceil \log_2 a(n) \rceil, n - a(n))$-BAPO solving* MA-JORITY *on length $n$ inputs.*

*Proof.* The BAPO is simple: $g$ attends to the first $n - a(n)$ tokens of the prefix and $f$ passes on the ones count in the remaining part of the prefix, which takes at most $\lceil \log_2 a(n) \rceil$ bits. This information suffices to allow the suffix oracle to solve the problem. $\qquad \square$

The hardness of MAJORITY immediately implies at least the same degree of hardness for MEDIAN, the problem of finding the median of a input sequence of integers, and MODE, the problem of finding the most frequent item in a stream. We believe these bounds are very loose.

**Corollary 2.** *No* $(o(\log n), O(n^{1-\epsilon}))$*-BAPOs solve* MEDIAN *or* MODE *on length $n$ inputs for any* $0 < \epsilon < 1$.

*Proof.* On bit-strings, the median and mode are 1 if and only if the string has a majority of 1's. $\quad \square$

For proving MATCH3$_n$ is BAPO-hard, the following lemma helps us find $x_i$–$x_j$ pairs that form a match with some particular suffix $s = x_n$, but such that $x_i$ and $x_j$ do not form matches with any suffix $s \in S$ and other prefix integer $z \in Z$, where $Z$ is the set of integers we have already decided to place in prefixes.

**Lemma 1.** *Let $S, Z \subset \mathbb{Z}_m$ with $m > 100$, $\max_{s \in S} s \leq \sqrt{m}$, $|S| \leq \sqrt{m}/2$, and $|Z| \leq \sqrt{m}/2$. For every $s \in S$, there exist $x, y \in \mathbb{Z}_m \setminus Z$ s.t.:*

1. $x + y + s \equiv_m 0$,

2. *for all $z \in Z$ and all $s' \in S$, $x + z + s' \not\equiv_m 0$ and $y + z + s' \not\equiv_m 0$.*

*Proof.* Condition 2 is satisfied as long as $x, y \in \mathbb{Z}_m \setminus Z$ are not in $-(S + Z) = \{-(s + z) \mod m : s \in S, z \in Z\}$, which has size at most $(\sqrt{m}/2)^2 = m/4$. This leaves at least $3m/4 - \sqrt{m}/2$ admissible values in $\mathbb{Z}_m \setminus Z$. We will show that there are enough $x$–$y$ pairs with distinct values of $x$ and $y$ satisfying Condition 1 that they cannot all be disallowed by Condition 2 and the requirement that $x, y \notin Z$.

Consider all pairs $x, y \in \mathbb{Z}_m$ such that $x + y + s \equiv_m 0$: for each $i \in \mathbb{Z}_m$, we can set $x = i \mod m$ and $y = -s - i \mod m$. If we consider $0 \leq i \leq m/4 + \sqrt{m}/2 + 1$, the values $x$ and $y$ take on are all distinct (since $s \leq \sqrt{m}$ and $m/4 > 2\sqrt{m}$, every $y$ value is larger than every $x$ value). Since there are only $m/4 + \sqrt{m}/2$ disallowed values for $x$ and $y$, there must be some $x, y$ pair in this range of more than $m/4 + \sqrt{m}/2$ values of $i$ where both $x$ and $y$ are admissible. $\qquad \square$

*Proof of Theorem 5.* First, consider MATCH2$_n$. The attention function $g$ can select any prefix elements $x_i$ such that $x_n + x_i \equiv 0 \pmod{m}$, since $x_n$ is always in the suffix. The suffix oracle only needs a single such example to confirm that this is a yes instance. If the set of attended tokens is empty, then the suffix oracle can check if there are any matches in the suffix. If not, this is a no instance. We do not need $f$ at all.

Now we show that MATCH3$_n$ is BAPO-hard. Suppose for a contradiction that $(f, g, h)$ is a BAPO solving MATCH3$_n$ with prefix bandwidth $o(n/b(n))$ and attention bandwidth $b(n)$. We will construct a collection of prefixes and suffixes that fools the BAPO with total input length $n$. Let $n > 10$ and pick $m = n^2$.

Consider the set of suffixes $S = \{i : 0 \leq i < n/(8b(n))\}$. Note than since $b(n) = o(n)$, $|S| = \Theta(n/b(n))$ is growing with $n$. For a suffix $s$, let $G(s) = \{(x, i) : i \in [n - 1], x \in \mathbb{Z}_{n^2}, g(s, n - 1, x_i, i) = 1\}$ be the set of integer–index pairs that $g$ attends to given suffix $s$.

1. Initialize $P^* = \{(\lfloor n^2/3 \rfloor), 1\}$ and $I = \{1\}$
   *We will use $\lfloor n^2/3 \rfloor$ as filler; placing it in $P^*$ here ensures it does not match any prefix integer.*

2. For $s \in S$:

   (a) While $|P^* \cap G(s)| < b(n)$:
      i. If there exists some $(x, i) \in \mathbb{Z}_{n^2} \times ([n - 1] \setminus I)$ for which (1) $(x, i) \in G(s)$ and (2) $x + y + s' \not\equiv_{n^2} 0$ for all $s' \in S$ and all $(y, j) \in P^*$: add $(x, i)$ to $P^*$ and add $i$ to $I$

> *Add a masking integer to the prefix, but only allow masking integers that do not form a match with existing masking integers and any suffix.*
>
> ii. Else: break out of the while loop
> *If there are no integers that $g$ wants to attend to (among those not forming matches with existing masking integers), then we are done with this suffix.*

After the procedure, $P^*$ contains at most $b(n)|S| + 1 \leq n/8 + 1$ occupied indices. Let $Z$ be the set of values in $P^*$, with $|Z| \leq n/8 + 1$. Moreover, we have ensured that no pair of integers in $P^*$ can form a match with any suffix integer $s$, since we only add $x$ to $P^*$ if for every $y$ in $P^*$ and $s \in S$, $x + y + s \not\equiv_{n^2} 0$. Lastly, if for some suffix $s$, we were unable to saturate attention and hit the else in 2(a)ii, then no other integers we add to a prefix can be attended to, since we will only be adding integers to prefixes that do not form matches with any integer in $P^*$ and any suffix (and therefore were already checked for attention in 2(a)i).

Now, for each $s \in S$, we find some $x_s$ and $y_s$ using Lemma 1 such that $x_s + y_s + s \equiv_{n^2} 0$, but $x_s$ and $y_s$ do not form matches with any other $z \in Z$ (recall that we have initialized $Z$ to contain all values in $P^*$). Add each $x_s$ and $y_s$ to $Z$ before the next application of Lemma 1 to ensure that we do not create any matches across $x$–$y$ pairs. After doing this for each $s \in S$, the final size of $Z$ is $\leq n/8 + 1 + n/(4b(n)) \leq 3n/8 + 1$, so the size limit on $Z$ required by Lemma 1, namely $|Z| \leq \sqrt{m}/2 = n/2$, is always satisfied (since we picked $n > 10$, $n/2 > 3n/8 + 1$). Let $P = \{(x_s, y_s)\}_{s \in S}$ be the $x$–$y$ pairs we find using this procedure.

For every subset $R \subseteq P$, construct a prefix of length $n - 1$ by first filling in all of the masking integers in $P^*$ (filling at most $n/8 + 1$ positions) and then adding in the $x$ and $y$ values in $R$ in arbitrary indices (filling at most $2|S| = n/(4b)$ additional positions). Fill the remaining indices with $\lfloor n^2/3 \rfloor$, which cannot form a match with itself and any suffix integer (since the suffix integers are at most $n/(4b(n))$), and which we already ensured cannot form a match with any integer in $P$ or $P^*$ by placing it in $Z$. This gives us $2^{|S|}$ distinct prefixes, each of which has matches with a distinct set of suffixes. That is, for any two prefixes $p_1 \neq p_2$, there exists some suffix integer $s$ where $p_1 s$ and $p_2 s$ have opposite answers to $\text{MATCH3}_n$, since there is some pair $(x_s, y_s)$ in one prefix, but not the other, which forms a match with $s$, while no other pair of integers forms a match with $s$. But with prefix bandwidth $o(n/b(n)) = o(|S|)$, there is some prefix oracle collision (as there are $2^{o(|S|)}$ distinct outputs of $f$, too few for the $2^{|S|}$ distinct prefixes). Moreover, these colliding prefixes are indistinguishable to attention for all suffixes, since we have ensured that for every suffix, attention can be saturated by integers in $P^*$, which are identical across all prefixes. Therefore the BAPO fails to solve the problem. $\qquad\square$

*Proof of Theorem 6.* For the $(2k, 0)$-BAPO, the prefix oracle transmits two bit-strings of length $k$, one indicating which elements of $\Sigma$ appear exactly once in the prefix and one indicating which elements of $\Sigma$ appear in the prefix one or more times. The suffix oracle outputs (a) any element that appears exactly once in the prefix but not in the suffix, which it finds from the first bit-string; (b) any element that appears exactly once in the suffix but not in the prefix, which it can find from the second bit-string; or (c) $\emptyset$ otherwise. This allows the suffix oracle to correctly solve UNIQUE.[3]

For the lower bound, suppose for a contradiction that $(f, g, h)$ is an $(o(k/b(k)), b(k))$-BAPO solving UNIQUE. We will first construct a set of partial prefixes $P^*$ for which the attention mechanism cannot distinguish all following suffixes. This leads to a pair of fooling prefixes with different answers that the BAPO cannot distinguish.

Fix an arbitrary order over the symbols in $\Sigma = \{\Sigma_1, \ldots, \Sigma_k\}$. For any $A \subseteq \Sigma$, let $\text{cat}(A)$ denote the string consisting of each token $\Sigma_i \in A$ concatenated in order. Let $b' = 4b(k)$. We will construct a collection of $k/b'$ suffixes (for convenience, assume $k/b'$ is an integer; otherwise, the construction would involve ceilings and floors, but this this would not affect the argument): let $\sigma_i = \text{cat}(\Sigma \setminus \{\Sigma_{b'(i-1)+j}\}_{j=1}^{b'})$ and $S = \{\sigma_i \sigma_i\}_{i=1}^{k/b'}$. As Figure 7 shows, the partial suffix $\sigma_i$ is missing the $i$th block of $b'$ contiguous tokens in $\Sigma$, so each full suffix $s_i = \sigma_i \sigma_i$ has length $2(k - b')$ and $|S| = k/b'$ by construction. Note that we have duplicated each $\sigma_i$ to ensure that no suffix token is unique.

---

[3]Additionally, there is a clever $(\lceil \log_2 |\Sigma| \rceil, 0)$-BAPO for the special case of UNIQUE where every element appears an even number of times, except a single unique item: with this restriction, taking the bit-wise exclusive or over binary encodings of the tokens solves the problem—but this fails in general.

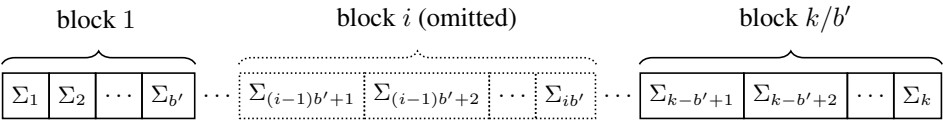

Figure 7: Constructing $\sigma_i$ by leaving out the $i$th block from the vocabulary $\Sigma$, leaving $k - b'$ tokens.

Now, we perform the usual overloading procedure to start building up prefixes of length $k$ that saturate the attention function for every suffix. Let $G(s) = \{(x, i) \in \Sigma \times [k] : g(s, k, x, i) = 1\}$. Perform the following procedure to construct a partial prefix $P^*$.

> Initialize $P^* \leftarrow \emptyset$, $I \leftarrow \emptyset$
> **for** $s \in S$ **do**
>     **while** $|P^* \cap G(s)| < b(k)$ **do**
>         **if** $\exists\, x \in \Sigma$, $i \in [k] \setminus I$ such that $(x, i) \in G(s)$ **then**
>             Add $(x, i)$ to $P^*$; add $i$ to $I$
>         **else**
>             **break**

This procedure results in at most $b(k)|S| = b(k)\frac{k}{b'} = b(k)\frac{k}{4b(k)} = k/4$ positions in $P^*$ being filled with masking symbols. Let $Z$ be the set of unique symbols in $P^*$, with $|Z| \leq |P^*| \leq k/4$. To ensure none of these are unique, fill in another $\leq k/4$ arbitrary indices in $P^*$ with an additional copy of each symbol in $Z$. As in the previous proofs, attention has now been rendered useless for every suffix, regardless of what additional symbols we add in remaining prefix indices. Consider the blocks we used to construct the suffixes, namely $\{\Sigma_{b'(i-1)+j}\}_{j=1}^{b'}$ for $i \in [k/b']$ (Figure 7). It must be the case that more than half of the blocks have an element not in $Z$ (suppose this was not the case and $> |S|/2$ blocks have all of their elements in $Z$, then $|Z| > b'|S|/2 = k/2$, contradicting that $|Z| \leq k/4$ ). Thus, we can find a collection of tokens $Y = \{y_1, \ldots, y_{|S|/2}\}$ such that each $y_i$ is in a different block and $Y \cap Z = \emptyset$, as well as one extra filler token $y_0$ from yet another different block.

For each subset $R \subseteq Y$, construct a prefix $p_R$ by starting with $P^*$, placing each $y_i \in R$ in an arbitrary unfilled index (so that the prefix now has $\leq k/2 + |S|/2$ filled indices) and filling the remaining indices with copies of $y_0$. This gives us $2^{|S|/2} = 2^{k/(2b')}$ distinct prefixes. With $o(k/b(k))$ prefix bandwidth, we only have $2^{o(k/b(k))}$ distinct outputs of $f$, so there are two prefixes $p_R$ and $p_{R'}$ that collide. These two prefixes have different subsets $R$ and $R'$ of $Y$ values, so there is some $y_i$ in one prefix but not the other (say $y_i \in R$, $y_i \notin R'$). Consider the suffix $s_{-i}$ that is missing the block of tokens to which $y_i$ belongs.

We observe that:

- $s_{-i}$ has no unique tokens due to its construction

- $[p_{R'}, s_{-i}]$ has no unique tokens, since all of its $Y$ values in the prefix appear in the suffix due to the fact that every element of $Y$ comes from a different suffix block. Thus, every element of $Y$ except $y_i$ appears in $s_{-i}$.

- $[p_R, s_{-i}]$ only has $y_i$ as a unique token, since (a) all other tokens in $Y$ have a copy in the suffix due to the above fact, (b) the masking and filler tokens in the prefix are all duplicated at least twice

Since the BAPO can observe the same attention set $G$ given both of these strings and receives the same output of $f$ from the two prefixes, it gets one of these instances wrong for some $G$.    □

*Proof of Theorem 7.* For the $(k, 0)$-BAPO, the prefix oracle sends a bit-string denoting all elements of $\Sigma$ appearing in $c$ but not $d$ (as far as it can tell). The suffix oracle can then remove all elements of $d$ in the suffix and compute an answer.

For the lower bound, as usual, suppose for a contradiction that $(f, g, h)$ is an $(o(k/b(k)), b(k))$-BAPO solving SETDIFF. We will use the a similar collection of omitted-block suffixes as we did for UNIQUE. As we did in that proof, order the elements of $\Sigma$ and define cat($A$). Let $b' = b(k)/4$. We

will construct a collection of $k/b'$ suffixes (again assuming for convenience that $k/b'$ is an integer): let $\sigma_i = \text{cat}(\Sigma \setminus \{\Sigma_{b'(i-1)+j}\}_{j=1}^{b'})$ and $S = \{\sigma_i\}_{i=1}^{k/b'}$ (recall Figure 7; note that in contrast with UNIQUE, there is no need to duplicate the suffixes).

Now, we perform the usual overloading procedure to start building up prefixes of length $3k + 1$ that saturate the attention function for every suffix. The prefixes we build up will be of the form $c_1 \ldots c_k | d_1 \cdots d_{2k}$. That is, the prefix–suffix split occurs somewhere in the middle of the $d$ input to SETDIFF. Let $G(\sigma) = \{(x,i) \in \Sigma \times [3k+1] : g(\sigma, k, x, i) = 1\}$. Perform the following procedure to construct a partial prefix $P^*$, which we initialize to already contain the divider token $|$.

> Initialize $P^* \leftarrow \{(|, k+1)\}$, $I \leftarrow \{k+1\}$
> **for** $\sigma \in S$ **do**
>     **while** $|P^* \cap G(\sigma)| < b(k)$ **do**
>         **if** $\exists\, x \in \Sigma, i \in [3k+1] \setminus I$ such that $(x,i) \in G(\sigma)$ **then**
>             Add $(x,i)$ to $P^*$; add $i$ to $I$
>         **else**
>             **break**

This procedure results in at most $b(k)|S| = b(k)\frac{k}{b'} = b(k)\frac{k}{4b(k)} = k/4$ positions in $P^*$ being filled with symbols that saturate $g$. Let $Z$ be the set of unique symbols in $P^*$, with $|Z| \leq |P^*| \leq k/4$. For each symbol in $Z$, place another copy of it in $P^*$ on the opposite side of the divider as its first copy. This ensures that no symbols in $Z$ can be SETDIFF answers, while leaving at least $3k/4$ open positions in $P^*$ before the divider and more than $k$ open positions after the divider.

Now, consider the blocks we used to construct the suffixes, namely $\{\Sigma_{b'(i-1)+j}\}_{j=1}^{b'}$ for $i \in [k/b']$ (Figure 7). Just as before, it must be the case that more than half of the blocks have an element not in $Z$. Thus, we can find a collection of tokens $Y = \{y_1, \ldots, y_{|S|/2}\}$ such that each $y_i$ is in a different block and $Y \cap Z = \emptyset$, as well as one extra filler token $y_0$ from yet another different block. Let $X = \Sigma \setminus (Z \cup Y)$. Place every $x \in X$ in an unfilled index of $P^*$ after the divider (i.e., in the $d$ portion) and fill all remaining empty spots of $P^*$ after the divider with the filler $y_0$.

For each subset $R \subseteq Y$, construct a prefix $p_R$ by starting with $P^*$ (which is now full after the divider, but has at least $3k/4$ open positions before the divider), placing each $r \in R$ in an arbitrary unfilled index before the divider, and then filling in all remaining indices with $y_0$. As before, this gives us $2^{|S|/2} = 2^{k/(2b')}$ distinct prefixes. With prefix bandwidth $o(k/b(k))$, we only have $2^{o(k/b(k))}$ distinct outputs of $f$, so there are two prefixes which have identical $f$ outputs, call them $p_R$ and $p_{R'}$. These contain different sets $R$ and $R'$ of $Y$ values; thus, there is some $y_i \in Y$ in one prefix but not the other. Suppose without loss of generality that $y_i \in R, y_i \notin R'$. Consider the suffix $\sigma_i$ that is missing the block of tokens to which $y_i$ belongs. Notice that:

- $p_R\sigma_i$ has only a single SETDIFF answer, which is $y_i$: we ensured that every $Y$ element only appears in a prefix before the divider, and $y_i$ does not appear in $\sigma_i$. Every other symbol in the missing block of $\sigma_i$ is in $p_R$ after the divider by construction, and thus is not a valid answer.

- $p_{R'}\sigma_i$ has SETDIFF answer $\emptyset$: the only symbol not appearing after the divider in $p_{R'}\sigma_i$ is $y_i$, which is not before the divider.

Thus these two instances have different SETDIFF answers, but they are indistinguishable to the BAPO (for some adversarially chosen $G$) due to the saturation of attention and the $f$ collision. $\qquad\square$

*Proof of Theorem 8.* Let $M = (Q, \{0,1\}, \Lambda, \delta, q_0, q_{\text{accept}}, q_{\text{reject}})$ be a Turing machine deciding $L$ using space $s(n)$.

A slight wrinkle arises due to the fact that BAPOs cannot attend to the last instance of a token, which would enable the efficient "tape diff" Turing machine simulation used by Merrill and Sabharwal [24]. As such, our construction requires a fixed maximum tape size, which means different BAPO-CoTs are needed for larger problem instances—but crucially, their bandwidths are identical. This is analogous to the requirement of Merrill and Sabharwal [24] that the precision of the transformer grows with the problem instance (although in our case, the scaling increases the number of chain-of-thought steps).

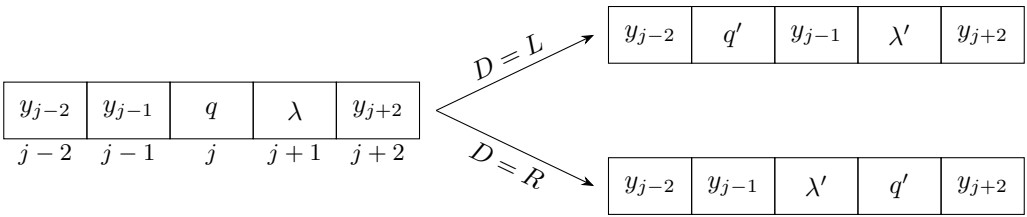

Figure 8: The tape head contents around the previous tape head's chunk offset after a step of the Turing machine $M$. All indices before $j - 1$ and after $j + 1$ are identical in the new chunk. For simplicity, the indices are shown relative to the start of the chunk.

Given $n$, we will construct a $(2, 3)$-BAPO-CoT that simulates $M$ on inputs of size at most $n$. Let $\Gamma = \Sigma \cup \Lambda \cup Q \cup \{\llcorner, \square\}$ be the token set for the BAPO-CoT. The BAPO-CoT will simulate $M$ by writing out the contents of the tape at each step of $M$, along with the current state, which will be written to the left of the tape cell where the tape head is currently positioned. Since only $s(n)$ tape cells are required, the BAPO-CoT will simulate a tape with exactly $s(n)$ cells. So, on input $x_1 \ldots x_n$, the first state the BAPO-CoT will write out is $q_0 x_1 \ldots x_n \llcorner\llcorner\llcorner \ldots$, with total length $c = s(n) + 1$, which we will call the chunk size. Let $\text{chunk}(i) = \lfloor i/c \rfloor$. We use $m$ to denote the current length of the BAPO-CoT's input (with $m = n$ at the first step) and $y = y_1, \ldots, y_m$ the current BAPO-CoT input itself (with $y_1 \ldots y_n = x$).

The prefix oracle $f$ is defined as follows:

$$f(y_1 \ldots y_k) = \begin{cases} 00 & \text{if the last symbol of } y_1 \ldots y_k \text{ is some } q \in Q \\ 01 & \text{if the symbol to the right of the last } q \in Q \text{ in } y_1 \ldots y_k \text{ is } 0 \\ 10 & \text{if the symbol to the right of the last } q \in Q \text{ in } y_1 \ldots y_k \text{ is } 1 \\ 11 & \text{otherwise } (y_1 \ldots y_k \text{ contains no symbols in } Q; \text{ every } q \text{ is followed by } 0 \text{ or } 1). \end{cases}$$

Thanks to this $f$, the suffix oracle always knows the symbol to the right of the state from the previous chunk. The attention function $g$ is defined as follows:

$$g(y_{k+1} \ldots y_m, k, y_i, i) = \begin{cases} 1 & \text{if } i = n - c \\ 1 & \text{if } i = n - c - 1 \\ 1 & \text{if chunk}(i) = \text{chunk}(m) - 1 \text{ and } y_i \in Q \\ 0 & \text{otherwise} \end{cases}$$

Thanks to this $g$, the suffix oracle always knows (a) the symbol at the current chunk offset index in the previous chunk, (b) the symbol before the one from (a), and (c) the state in the previous chunk and its chunk offset index (i.e., the tape head position). If any of these positions are in the suffix, they are directly observed by the suffix oracle and if they are in the prefix, then they are contained in the attended set $G$.

The suffix oracle $h$ performs the following procedure given $f(y_1 \ldots y_k)$, $G$, $k$, and $y_{k+1} \ldots y_m$:

1. If $\text{chunk}(m) = 0$, return $\llcorner$

2. If $\text{chunk}(m) = 1$:

    (a) If $m = c + 1$: return $q_0$
    (b) Else: return $y_{m-c-1}$ ($y_{m-c-1}$ is either in the suffix or in $G$)

3. Else: let $i = m \mod c$. Let $j$ be the head position in the previous chunk and let $q$ be the state in the previous chunk. Note that $q$, $j$, $y_{m-c}$, and $y_{m-c-1}$ are all known to $h$, since they are either included in the suffix or in $G$ ($j$ is computable from the positional encoding of $q$). Moreover, $\lambda = y_{\text{chunk}(m-c) \cdot c + j + 1}$ (the symbol under the tape head in the previous chunk) is either in the suffix or can be inferred given the suffix and $f(y_1 \ldots y_k)$. The suffix oracle can thus compute the step of the Turing machine $\delta(q, \lambda) = (q', \lambda', D)$, where $q' \in Q$ is the next state, $\lambda'$ is the symbol written to the tape, and $D \in \{L, R\}$ is the direction the tape head moves.

First, we check to see if $M$ has halted. If $q' = q_{\text{accept}}$: if $y_m$ is not 1, return 1, otherwise return □. If $q' = q_{\text{reject}}$: if $y_m$ is not 0, return 0, otherwise return □. This ensures we output the answer and then terminate. Otherwise, we proceed to simulate $M$. To output the next symbol given a step of the Turing machine $M$ applied to the previous chunk (see Figure 8):

(a) If $i = j - 1$: if $D = L$, return $q'$; if $D = R$, return $y_{m-c}$
(b) Else if $i = j$: if $D = L$, return $y_{m-c-1}$; if $D = R$, return $\lambda'$
(c) Else if $i = j + 1$: if $D = L$, return $\lambda'$; if $D = R$, return $q'$
(d) Else: return $y_{m-c}$

This procedure results in the updated tape of $M$ being written out symbol-by-symbol into the next chunk. Since $M$ decides $L$, it will eventually halt, at which point the above BAPO-CoT outputs the answer (0 or 1) and then outputs □ to terminate. □

## C  BAPO generalizations

We consider three generalizations of our model, *score-BAPO*, *multi-layer BAPO*, and *full-attention BAPO*. We show that these variants (even in combination) do not affect the BAPO-hardness of REACHABILITY, MAJORITY, or MATCH3$_n$. However, they do enable solving the (multi-hop) induction heads task, which appears hard under the original BAPO definition. This suggests multiple degrees of hardness, with our set of three BAPO-hard tasks appearing fundamentally hard even as we make the model align with more details of the transformer architecture.

### C.1  Score-BAPO

A score-BAPO has an attention function that outputs a real number in $[0, 1]$ rather than only 0 or 1. The attended set $G$ then consists of the (up to) $b$ tokens with highest non-zero attention scores (if there are multiple such sets, the score-BAPO must be robust to an arbitrary attended set, as with the original BAPO). This is a strict generalization, as a score attention function can simulate a binary one by outputting scores 0 or 1.

**Definition 11.** An $(a, b)$-score-BAPO is defined by a *prefix oracle* $f : \Sigma^* \to \{0, 1\}^a$, an *attention function* $g : \Sigma^* \times \mathbb{N} \times \Sigma \times \mathbb{N} \to [0, 1]$, and a *suffix oracle* $h : \{0, 1\}^a \times \bigcup_{i=0}^{b} (\Sigma \times \mathbb{N})^i \times \Sigma^* \times \mathbb{N} \to \Sigma$. An $(a, b)$-score-BAPO *solves* a computational problem $p : \Sigma^* \to \Sigma$ if $h(f(x_1 \ldots x_k), G, x_{k+1} \ldots x_n, k) = p(x_1 \ldots x_n)$ for all $k < n$ and all $G \in \arg\max_{\substack{S \subseteq \{(x_i, i): i \le k, g_i > 0\} \\ |S| \le b}} \sum_{(x_i, i) \in S} g_i$, where $g_i = g(x_{k+1} \ldots x_n, k, x_i, i)$.

A constant-bandwidth score-BAPO can solve some problems that otherwise appear to be BAPO-$\Sigma$-hard or possibly even BAPO-hard. Two problems that a score-BAPO can solve that seem hard for BAPOs are MAX and RIGHTMOST (and their symmetric variants), defined as follows.

**Definition 12.** MAX: $\mathbb{Z}_m^* \to \mathbb{Z}_m$ is the problem of finding the maximum of a list of integers between 0 and $m - 1$. MIN is defined analogously.

**Definition 13.** RIGHTMOST: $\Sigma^* \times \Sigma \to \mathbb{N} \cup \{-1\}$ is the problem of finding the index of the rightmost instance of a token in a list, or $-1$ if the item is not in the list. The token to search for is provided in the input after the list. That is, RIGHTMOST$(x_1 \ldots x_{n-1}, x_n) = \max(\{-1\} \cup \{i \in [n-1] : x_i = x_n\})$. LEFTMOST is defined analogously.

**Proposition 2.** *There are $(0, 1)$-score-BAPOs for* MAX*,* MIN*,* LEFTMOST*, and* RIGHTMOST*.*

*Proof.* For MAX, the attention function $g(x_{k+1} \ldots x_n, k, x_i, i) = x_i/m$ ensures $G$ contains the largest element in the prefix (or $G = \emptyset$ if the prefix is all 0s), which $h$ can use in combination with the suffix to identify the largest element of the sequence. The attention function $g(x_{k+1} \ldots x_n, k, x_i, i) = (m - x_i)/m$ performs the same function for MIN. For RIGHTMOST, the attention function is:

$$g(x_{k+1} \ldots x_n, k, x_i, i) = \begin{cases} i/k & \text{if } x_i = x_n \\ 0 & \text{otherwise} \end{cases}$$

This ensures $G$ contains the rightmost occurrence of $x_n$ in the prefix (or $G = \emptyset$ if $x_n$ does not occur in the prefix), which, combined with the suffix, allows $h$ to solve the problem. Using $(k - i + 1)/k$ instead of $i/k$ provides a solution to LEFTMOST. $\qquad\square$

## C.2 Multi-layer BAPO

A $d$-layer BAPO has a constant number of attention functions $g_1, \ldots, g_d$ that operate sequentially, with each having access to the attended tokens from the previous layer. The suffix oracle $h$ then solves the problem using the last attended set $G_d$. This allows a $d$-layer BAPO to perform multi-hop tasks as a multi-layer transformer would. The original BAPO model is just a 1-layer BAPO, and a BAPO with any number of layers can simulate one with fewer layers by having some initial number of attention functions act as no-ops. Note that the total number of attended tokens $h$ has access to in an $(a, b)$ $d$-layer BAPO is $bd$, still a constant when $b = O(1)$ and $d = O(1)$.

**Definition 14.** An $(a, b)$ $d$-layer BAPO is defined by a *prefix oracle* $f : \Sigma^* \to \{0,1\}^a$, $d$ *attention functions* $g_i : \Sigma^* \times \mathbb{N} \times \Sigma \times \mathbb{N} \times \cup_{j=0}^{b}(\Sigma \times \mathbb{N})^j \to \{0,1\}$ for $i = 1, \ldots, d$, and a *suffix oracle* $h : \{0,1\}^a \times \cup_{i=0}^{b}(\Sigma \times \mathbb{N})^i \times \Sigma^* \times \mathbb{N} \to \Sigma$. Given an instance $x_1 \ldots x_n \in \Sigma^n$ of a computational problem $p : \Sigma^* \to \Sigma$ and a split index $k < n$, an *attended set sequence* $G_1, \ldots, G_d$ is some sequence of sets where $G_1 \subseteq \mathbb{G}_1 = \{(x_i, i) : 1 \le i \le k, g_1(x_{k+1} \ldots x_n, k, x_i, i, \emptyset) = 1\}$ and $G_{j+1} \subseteq \mathbb{G}_{j+1} = \{(x_i, i) : 1 \le i \le k, g_{j+1}(x_{k+1} \ldots x_n, k, x_i, i, G_j) = 1\}$, with each $|G_j| = \min\{b, |\mathbb{G}_j|\}$ for $j = 1, \ldots, d - 1$. An $(a, b)$-$d$-layer BAPO *solves* a computational problem $p : \Sigma^* \to \Sigma$ if $h(f(x_1 \ldots x_k), G_d, x_{k+1} \ldots x_n, k) = p(x_1 \ldots x_n)$ for all $k < n$ and all attended set sequences $G_1, \ldots, G_d$.

## C.3 Full-attention BAPO

A *full-attention BAPO* is a minor variant where the attention function $g$ may attend to all tokens in the input, not just those in the prefix. That is, we now use $\mathbb{G} = \{(x_i, i) : 1 \le i \le n, g(x_{k+1} \ldots x_n, k, x_i, i) = 1\}$ (rather than $1 \le i \le k$), with no other changes. A full-attention BAPO can act just like a standard BAPO by having $g(x_{k+1} \ldots x_n, k, x_i, i) = 0$ whenever $i > k$. This modification provides no increase in expressive power, as $h$ has access to all suffix tokens anyway, and is a pure convenience measure for combining with multi-layer BAPO.

## C.4 Combining variants to solve the induction heads task

These extensions can be combined. For instance, we can define a *$d$-layer full-attention score-BAPO*, where each of $d$ attention functions outputs a score over all input tokens. We show that this BAPO variant can solve the multi-hop induction heads task [37], while still leaving REACHABILITY, MAJORITY, and MATCH3$_n$ BAPO-hard. (This also implies that each of the three variants alone leaves these problems hard, as the capabilities enabled by any of the variants can be ignored.) The requirement of a second layer of processing for one-hop induction heads is analogous to known results that the problem is efficiently solvable by two-layer but not one-layer transformers [7, 36].

**Definition 15.** $d$-HOP INDUCTION HEADS: $\Sigma^* \to \Sigma \cup \{\bot\}$ is the problem of iteratively finding the token that follows the rightmost previous occurrence of the last token. Formally, the solution on input $x_1 \ldots x_n$ is given by $x_{\mathsf{hop}_k(x)}$ if $\mathsf{hop}_k(x) \ne 0$ else $\bot$, where

$$\mathsf{hop}_1(x) = \max(\{0\} \cup \{j \in [n] : x_{j-1} = x_n\})$$
$$\mathsf{hop}_{i+1}(x) = \max(\{0\} \cup \{j \in [\mathsf{hop}_i(x)] : x_{j-1} = x_{\mathsf{hop}_i(x)}\}). \qquad \text{(for } i = 1, \ldots, k - 1\text{)}$$

**Theorem 9.** *For any $d \ge 1$, there is a $(0, 1)$-$2d$-layer full-attention score-BAPO for $d$-HOP INDUCTION HEADS.*

*Proof.* The idea behind the construction is that the $2d$ attention functions alternate between attending to the rightmost instance of the current token being searched for (i.e., attending to the rightmost previous instance of the token at index $\mathsf{hop}_j(x) - 1$) and attending to the token to the right of that rightmost instance (i.e., attending to the index $\mathsf{hop}_j(x)$).

More precisely, each $g_j$ is defined as follows:

1. For $j = 1$, the first attention layer finds the rightmost previous occurrence of the last token, using:

$$g_1(x_{k+1} \ldots x_n, k, x_i, i, \emptyset) = \begin{cases} i/n & \text{if } i < n \text{ and } x_i = x_n \\ 0 & \text{otherwise} \end{cases}$$

This ensures $G_1$ contains the rightmost instance of the token $x_n$ (or $\emptyset$ if $x_n$ does not appear earlier in the input). Note that since this is a full-attention BAPO, $G$ contains this rightmost instance even if it appears in the suffix. (How convenient!) Using induction heads notation, $G_1$ contains the token at index $\mathsf{hop}_1(x_1 \ldots x_n) - 1$ (or $G_1 = \emptyset$ if $\mathsf{hop}_1(x_1 \ldots x_n) = \bot$).

2. For even $j$, we have the invariant that $G_{j-1}$ (provided to $g_j$) contains the rightmost instance of the current token being searched for (or $\emptyset$ if no instance of the token was found). (When defining $g_j$ for larger odd $j$, we will maintain this invariant.) We then define $g_j$ as follows:

$$g_j(x_{k+1} \ldots x_n, k, x_i, i, G_{j-1}) = \begin{cases} 1 & \text{if } G_{j-1} = \{(x_\ell, \ell)\} \text{ and } i = \ell + 1 \\ 0 & \text{otherwise} \end{cases}$$

This ensures $G_j$ contains the token appearing to the right of the one identified in the previous layer (or $\emptyset$ if the induction heads chain has been broken). Using the notation of induction heads, $G_j$ contains the token at index $\mathsf{hop}_{j/2}(x_1 \ldots x_n)$ (or $G_j = \emptyset$ if $\mathsf{hop}_{j/2}(x_1 \ldots x_n) = \bot$).

3. For odd $j > 1$, the above definition ensures $G_{j-1}$ contains the token at index $\mathsf{hop}_{(j-1)/2}(x_1 \ldots x_n)$ (or is $\emptyset$). The next attention function $g_j$ thus needs to look for the rightmost earlier occurrence of that token, which is accomplished by:

$$g_j(x_{k+1} \ldots x_n, k, x_i, i, \emptyset) = \begin{cases} i/n & \text{if } G_{j-1} = \{(x_\ell, \ell)\}, i < \ell, \text{ and } x_i = x_\ell \\ 0 & \text{otherwise} \end{cases}$$

This ensures $G_j$ contains the token at index $\mathsf{hop}_{j/2}(x_1 \ldots x_n) - 1$, or is empty if no such token exists (or $G_{j-1}$ was empty).

As each layer of attention maintains the needed invariants for the next layer, an inductive argument shows that $G_{2d}$ contains the token at index $\mathsf{hop}_d(x_1 \ldots x_n)$ (or is empty if the chain was broken at any point). Thus $h$ can return this token (or $\bot$) and solve $d$-HOP INDUCTION HEADS. $\square$

## C.5 Our BAPO-hardness proofs are robust to these variants

Given these variants and their ability to solve problems that appear difficult for the standard BAPO model, one question is whether the model is too powerful and trivializes our hardness results. However, we show that REACHABILITY, MAJORITY, and MATCH3$_n$ remain hard for a $d$-layer full-attention score-BAPO with constant $d$. The proof technique we use for BAPO-hardness extends naturally to these variants, so any problem shown to be BAPO-hard using our approach is also hard under these variants.

**Theorem 10.** *For any $d = O(1)$, there is no constant-bandwidth $d$-layer full-attention score-BAPO for* REACHABILITY, MAJORITY, *or* MATCH3$_n$.

*Proof.* The proof structure and constructions remain the same as in Theorems 3 to 5; we only need to change the way masking tokens are selected to account for the new structure of the attention functions (and we will need to use more masking tokens as the effective number of tokens attended to is $bd$ rather than $b$, but this is only a constant factor). The idea is that when placing masking tokens in the shared prefixes, we need to iterate through each attention function, masking the attention of each one in turn. In this iterative process, we use the constructed attended token set containing only masked tokens as the input to the next attention function. Additionally, at each step, we select the masking tokens that have the $b$ highest attention scores (rather than an arbitrary set of attended tokens, as we did in the binary attention case). The fact that suffix tokens may be attended to only helps, as this can never help $h$ distinguish between fooling instances (as they share the same suffix).

We provide the updated prefix construction for REACHABILITY as an example (the other two are analogous transformations of the original proofs). Refer to the proof of Theorem 3 for full notation.

We need a few additional definitions for the proof extension. Let $b^{\text{th}}(r, s_i t_j, P^*)$ be the $b^{\text{th}}$ largest attention score that $g_r$ outputs on any edge in $P^*$ (positioned at their canonical indices given by $I(\cdot)$) or on the suffix tokens $s_i$ and $t_j$. If there are fewer than $b$ candidate tokens between $P^*$ and the suffix, then $b^{\text{th}}(r, s_i t_j, P^*) = 0$. Given a non-target node $u$, let $\mathsf{next}(u, P^*)$ be the set of nodes in the layer to the right of $u$ that have in-degree 0 in $P^*$.

1. Initialize $P^*$ to have the same set of nodes as $P$. Initialize $S$ to be the set of all $s_i$ and $u_{ij}$ nodes, which will store the current set of nodes that still need an outgoing edge.

2. For each pair $(i, j) \in [p] \times [p]$:

   (a) Initialize $G_0 = \emptyset$
   (b) For $r = 1, \dots d$:
      i. While $\max_{u \in S, v \in \mathsf{next}(u, P^*)} g_r(s_i t_j, m, (u, v), G_{r-1}, I(u)) > b^{\text{th}}(r, s_i t_j, P^*)$: add the maximizing edge $(u, v)$ to $P^*$ and remove $u$ from $S$.
      *Ensure the top $b$ still-feasible edges that have highest attention scores under $g_r$ are in $P^*$; thus, no edge we add later will have a higher attention score for $g_r$ on suffix $s_i t_j$.*
      ii. Let $G_r$ be a set of $b$ tokens in $P^*$ (and $s_i t_j$, since we are using full attention) with highest score under $g_r$ on suffix $s_i t_j$. If there are fewer than $b$ such tokens with non-zero score, take all tokens with non-zero score.
      *This $G_r$ is one possible attended set at this step, regardless of how the prefix is completed (or, without attention score ties, the unique attended set), since we have ensured the $b$ highest-scoring feasible edges are added to $P^*$. For adding masking tokens in the next attention layer, we will assume this is the attended set.*

3. Let $E^*$ be the set of edges in $P^*$ at this point in the algorithm. These will be shared among all prefixes to saturate attention. To complete the paths in $P^*$ arbitrarily, connect each node with out-degree 0 (that is not a $t$ node) to the first node with in-degree 0 in the next layer.

The total number of edges we add to $P^*$ is at most $bdp^2$ (compared to $bp^2$ in the original construction). Thus, as long as $bd = o(m^{1-2/c})$ (which is certainly true with $b = O(1)$ and $d = O(1)$), we have enough edges in the graph to mask attention for every layer and for every suffix. Given these masking edges, we have found some sequence of attended sets $G_1, \dots, G_d$ (those used in the construction) for every suffix that only contain tokens shared among all prefixes, so the rest of the proof remains the same. On a pair of fooling instances as described in the original proof, the suffix oracle sees the same final attended set $G_d$ regardless of the prefix, the same output of $f$, and the same suffix—thus making a mistake.

The same approach (picking the masking tokens with the $b$ highest values of $g_r$ for each $r = 1, \dots, d$) allows us to generalize the BAPO-hardness proofs of MAJORITY and MATCH3$_n$ with only an additional constant factor $d$ of masking tokens. Since this only requires a constant factor of additional masking tokens, any BAPO-hardness proof using the same structure can be extended to $d$-layer full-attention score-BAPO. $\qquad\square$

## D   Experiments

### D.1   Implementation Details

All models were forced to output a pre-set JSON schema, shown in the tables below. The model versions and API settings were as follows:

| Family | Model | Version Specifier | Temperature | Other Params |
|--------|-------|-------------------|-------------|--------------|
| GPT | 4o | `gpt-4o-2024-11-20` | 0 | |
| | 4o mini | `gpt-4o-mini-2024-07-18` | 0 | |

| | | | | |
|---|---|---|---|---|
| o3 | | o3-2025-04-16 | n/a | {effort: medium} |
| Claude | 3.5 Sonnet | claude-3-5-sonnet-20241022 | 0 | |
| | 3.5 Haiku | claude-3-5-haiku-20241022 | 0 | |
| Gemini | 1.5 Pro | gemini-1.5-pro-002 | 0 | |
| | 1.5 Flash | gemini-1.5-flash-002 | 0 | |
| | 2.5 Flash | gemini-2.5-flash-preview-04-17 | 0 | |

The experiments took $\leq$ 1 day and ~$400 of API credits to run ($93 of which were for o3 alone), with preliminary experiments taking an additional ~$150 of API credits.

### D.1.1 Base Experiments

Below are the instructions used for each task, illustrated using one example instance:

| Experiment | Example Prompt | Example Output |
|---|---|---|
| INDEX | Output the element at the specified index (starting at 0) of the list: List: [{"index": 0, "value": 117}, {"index": 1, "value": 30}, {"index": 2, "value": 169}, {"index": 3, "value": 113}, {"index": 4, "value": 52}, {"index": 5, "value": 168}] Index: 0 | {"element_value": 117} |
| EQUALITY | Output true if the left and right lists are identical: Left: [1, 1, 0, 1, 0, 1, 1, 0, 0, 1] Right: [0, 1, 0, 1, 0, 1, 1, 0, 1, 1] | {"equals": false} |
| MATCH2 | You are given a list of numbers and a number x. Determine whether list[i] + x = 0 for some i. List: [-300, 62, 144, -490, 469] x: -144 | {"found_i": true} |
| REACHABILITY | You are given an directed graph with 6 nodes as a list of edges (i, j). An edge (i,j) means that node i points to j. The edges in G are: [[5, 4], [2, 0], [4, 1], [3, 2]] Is there a path from node 5 to node 1? | {"path_exists": true} |
| MAJORITY | Output true if the majority of elements of this list are 1, else false: [0, 1, 0, 0, 1, 1, 1] | {"majority_is_1s": true} |
| MATCH3 | You are given a list of numbers and a number x. Determine whether list[i] + list[j] + x = 0 for some i, j. List: [508, 567, -178, 382, -240] x: -890 | {"found_i_and_j": true} |

| | | |
|---|---|---|
| DISJOINTNESS | These left and right lists represent sets using binary indicators for each item. Output true if these sets are disjoint and false if they have a non-empty intersection. That is, output true if and only if there is no index where both lists contain 1. Left: [0, 1, 1, 1, 0, 1] Right: [0, 0, 0, 0, 1, 0] | {"is_disjoint": true} |
| INTDISJOINTNESS | Output true if left and right lists are disjoint (share no elements) and false otherwise: Left: [73, 290, 133, 342, 142, 279] Right: [236, 16, 306, 144, 279, 242] | {"is_disjoint": false} |
| UNIQUE | Output the element in the list that occurs only once: [4, 4, 4, 4, 6] | {"unique": 6} |
| SETDIFF | You are given two sets of numbers A and B. Output the element in set A that is not in set B. If there is no such element, output -1. Set A: [3, 5] Set B: [5, 2, 3] | {"element": -1} |

To generate the problem instances, we used the procedures below. We use a grid of $n \in \{6, 50, 100, 200\}$ but resulting list lengths might deviate slightly if a problem requires odd numbers. We generated an equal number and positive and negative instances where applicable.

- INDEX: For each run, sample a permutation $\pi$ of $\{0, \ldots, 199\}$; set $x = \pi_{1:n-1}$, choose $i \sim \text{Unif}(0, n-1)$.
- EQUALITY: Sample $x \in \{0, 1\}^n$ uniformly. Let $y = x$ for positives. For negatives, choose $i \neq j$ with $x_i \neq x_j$, swap $y_i, y_j$.
- MATCH2: Generate permutation $\pi \sim \text{Perm}(0, 999)$. Set $x = \pi_{1:n}$. Then, to generate
  - Positives: inject $-x_n$ into $x_{1:n-1}$ at a random position
  - Negatives: ensure $-x_n \notin x_{1:n}$ by replacing it with a random element from $\pi$.
- REACHABILITY: Construct $k = 2$ node-disjoint paths of length $\ell = n/k-1$. Let graph $G$ be union of paths, map node names to integers via random bijection $\sigma$. Choose $(s, t)$ such that:
  - Positives: $s, t$ on same path $\Rightarrow$ path exists
  - Negatives: $s, t$ on different paths $\Rightarrow$ no path
- MAJORITY: Generate $x \in \{0, 1\}^{n+1}$ such that majority bit occurs $\lceil (n+1)/2 \rceil$ times. Shuffle $x$.
- MATCH3: Sample $\pi \sim \text{Perm}(-250, 1000)$. Reject if $x = \pi_{1:n}$ has a triplet that satisfies $x_i + x_j + x_n = 0$ (negative). For positives, generate negative instance and then select $i \neq j$, set $x_n = -(x_i + x_j)$.
- DISJOINTNESS: Sample $(x_i, y_i)$ uniformly from $\{(0, 0), (0, 1), (1, 0)\}$.
  - Positives: already disjoint
  - Negatives: pick $j$, set $a_j = b_j = 1$
- INTDISJOINTNESS: This is slightly different from DISJOINTNESS to avoid shortcuts from varying set sizes. Generate permutation $\pi \sim \text{Perm}(0, 399)$. Set $x = \pi_{1:n/2}$ and $y = \pi_{n/2:n}$. For negatives, set $x_j = y_j$ for random index $j$.

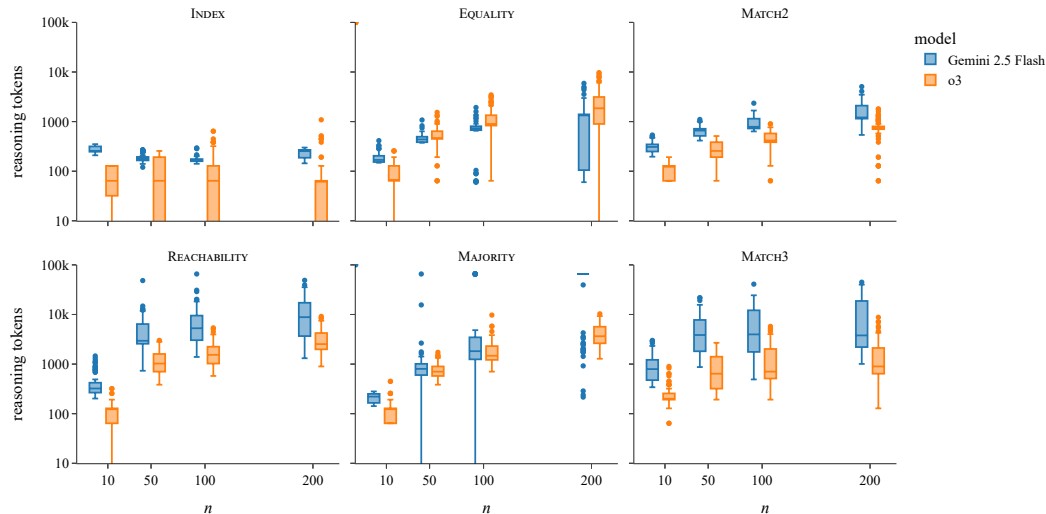

Figure 9: Number of reasoning tokens used by o3 and Gemini 2.5 Flash for each problem. The models perform well on BAPO-hard tasks in line with our BAPO-CoT result, but they use thousands or even tens of thousands of CoT tokens to do so.

- UNIQUE: Generate permutation $\pi \sim \mathrm{Perm}(0, n-1)$. Then, place one unique element $u$ and fill remaining with elements of frequency $\geq 2$ by drawing from $\pi_{1:n/4}$.
- SETDIFF: Generate permutation $\pi \sim \mathrm{Perm}(0, n-1)$. Sample unique element $u$. Split $\pi$ into parts and recombine:

$$A = S \cup \{u\}, \quad B_{pos} = S \cup V \cup \{u\}, \quad B_{neg} = S \cup V \cup \{v\}$$

where $S$ = shared, $V$ = unique-to-B, $v$ = randomly sampled replacement element from $S$. $|S| = |V| + 1$.

### D.1.2 CoT Experiments

For the chain of thought variants, we pre-pended the following instructions:

```
Think step by step on the CoT, but stay under 250 words.
```

The output JSON object then contained a `cot` field before the actual answer, e.g.,

```
{"cot": "To determine if there exist indices i and j ...", "found_i_and_j": true}
```

Additionally, we ran experiments with two reasoning models that also supported structured outputs, OpenAI's o3 and Google's Gemini Flash 2.5. Model version and parameters can be found above. Figure 9 shows the number of reasoning tokens the models used.

### D.1.3 Real-World Experiments

| Experiment | Example Prompt | Example Output |
|---|---|---|

| | | |
|---|---|---|
| VARIABLETRACKING | In the Python code below, is x7 == "a" at the end of execution?
```python
x6 = "a"
x4 = "b"
x0 = x6
x2 = x4
x3 = x0
x8 = x2
x9 = x3
x7 = x3
x1 = x8
x5 = x8
```| {"is_equal": true} |
| MAJORITYREVIEW | Output true if the majority of the following reviews is positive, else false.
[ { "id": 0, "review": "I loved the grand entrance hall with its two impressive chandeliers and a player grand piano. I took advantage of the exercise room in the basement and loved getting a coffee from the bar to take up to my room. The cleaning staff were particularly pleasant, greeting me every time we happened to pass. The lift [...] | {"majority_is_positive": true} |
| FINDNEGATIVEREVIEW | Return the id of the most negative review.
[ { "id": 0, "review": "Lovely hotel and great location. I can recommend this hotel, the location is great for all tourist attractions and the airport. Very friendly staff and worth every penny." }, { "id": 1, "review": "We stayed at the Boston Park Plaza Hotel in April and couldn't have been happier. The hotel is centrally located near the Public Gardens, Theater District and Boston Commons which [...] | {"most_negative": 8} |

**Dataset Processing Details.** We used hotel reviews from the SPACE dataset[4] [2]. Reviews with a rating of 5 get a positive label, reviews with a rating of 1 get a negative label to ensure a clear separa-

[4]Available at `https://github.com/stangelid/qt` under an MIT License.

tion between classes. We annotate the resulting reviews with GPT4.1-nano to check for consistency and only keep the ones where the original label agrees with the LLM annotation. Finally, we only keep hotels with at least 101 positive and 53 negative labels to make sure we have a large enough set to subsample from.

- VARIABLETRACKING: Follow the same process as for REACHABILITY to construct a graph with $k = 2$ paths. Let these two paths be $p$ and $p'$.
  - Then choose $i, j$ with $j < i$ and insert a cross path edge from $p_j \to p'_i$, erasing $p'_{i-1} \to p'_i$.
  - Map nodes to variable names $x_0, x_1, \ldots$ via random bijection and initialize all nodes with no incoming edges to a letter from the alphabet.
  - Choose $s$ and $t$ as with REACHABILITY
  - Finally, generate assignment statements via sampling a random topological ordering of the overall graph.

- MAJORITYREVIEW: For each label $y \in \{\text{True}, \text{False}\}$, sample $n/2 + s$ reviews of label $y$, and $n/2$ of the opposite label, where $s = 3$ is a slack variable to help reduce any remaining noise in the reviews. Shuffle the combined list of reviews.

- FINDNEGATIVEREVIEW: We iterate through hotels in round-robin fashion. Sample $n - 1$ positive and 1 negative review to generate data for one task instance. Shuffle those in random order.

### D.2 Additional Results

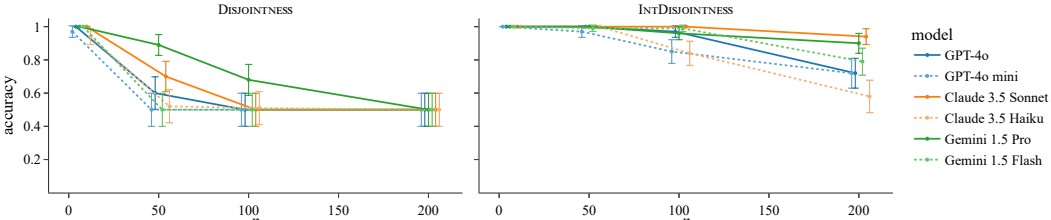

Figure 10: Additional results on BAPO-easy problems. INTDISJOINTNESS is a variant of DISJOINTNESS where sets are represented by the indices of elements they contain instead of binary vectors to show that positional encodings rather than BAPO-hardness are likely to be responsible for the poor performance on this task.

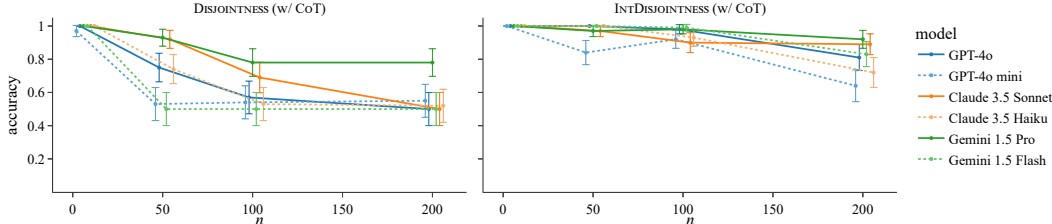

Figure 11: Adding CoT to BAPO-easy problems provides a boost to larger models on DISJOINTNESS, especially Gemini 1.5 Pro, while performance remains high on the other problems.

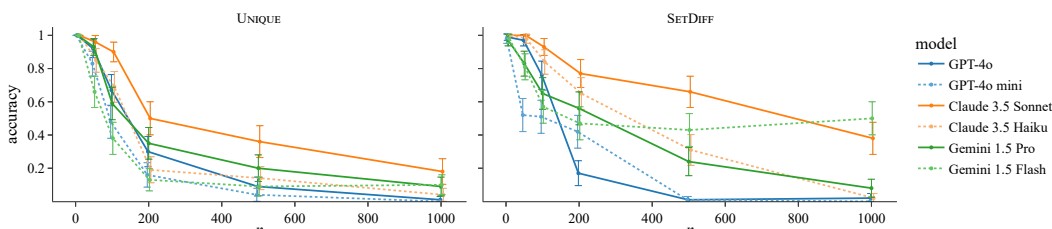

Figure 12: The hardness of UNIQUE and SETDIFF scales with the vocabulary size which we try to increase via input length $n$ here. Drops still occur, but appear to be less pronounced, perhaps because pre-trained LLMs have fixed token representations and scaling input length is only a proxy for increased vocabulary size.

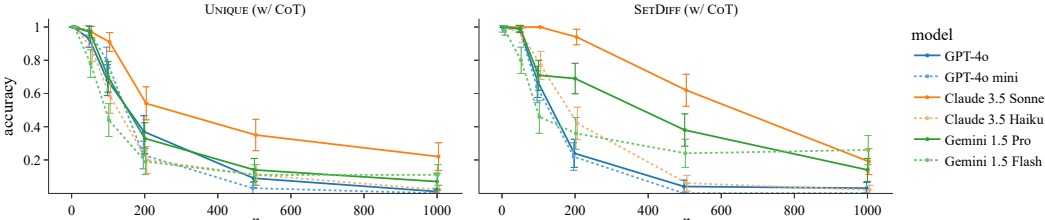

Figure 13: Adding CoT to BAPO-$\Sigma$-hard problems does not result in substantial changes. We conjecture that this is again due to the static nature of the underlying vocabulary representations.

