# OpenReview forum: "Lost in Transmission: When and Why LLMs Fail to Reason Globally"
_NeurIPS.cc/2025/Conference — NeurIPS 2025 spotlight_

### Official Review · Reviewer_qUKG · 2025-06-03

**Clarity:** 3
**Significance:** 3
**Originality:** 2
**Rating:** 5
**Confidence:** 2

**Summary:**

Reasoning across the global input context is important to boosting accurate task performance. This paper introduces a theoretical model to bound the communication constraints due to causal attention in transformer-based large language models (LLMs): bounded attention prefix oracle (BAPO). The authors use the framework to outline 3 classes of problems: BAPO-easy, BAPO-hard, BAPO-$\sum$-hard. They provide upper and lower bounds on the (prefix, attention) bandwidth. They then empirically show that these hardness classes are indicative of actual LLM performance on the analyzed problems. They further extend the empirical results to include real-world problems in the area of sentiment analysis and code tracing.

**Questions:**

- Can you include the theoretical lower and upper limits of BAPOs with COT similar to what was provided in table 1 for BAPO with no COT?
- You only consider production-grade language models in your empirical analysis. At what point during model training do LLMs become well trained enough to fit the BAPO framework. Presumably, randomly initialized LLMs would not perform well on any task.
- I am curious to see if adding in-context examples to the real-world tasks, as is suggested in the discussion (lines 324-330), empirically improves performance?

**Ethical Concerns:**

["NO or VERY MINOR ethics concerns only"]

**Final Justification:**

- The strengths outlined in my initial review outweighed the weaknesses.
- In the discussion period, the authors were able to clarify a misunderstanding about when BAPO framework can be applied to a model.
- Therefore, I maintain my positive score.

**Limitations:**

yes

**Paper Formatting Concerns:**

line 289: "requiremd" --> typo

**Quality:**

3

**Strengths And Weaknesses:**

**Quality:**

The theoretical model proposed (BAPO) is both analyzed via proof and empirical experiments which support the authors main arguments.

**Clarity:**

The paper is well organized, and the experiments are well motivated by the theoretical analysis.

Weakness:
- In figures 3, 4, and 5, the theoretical lower and upper bound should be included to allow for quick comparison.
- When trying to compare figure 3 vs. 4 I found myself having to quickly flip between the two figures to compare w/ and w/o COT. It would be helpful if you could summarize the COT vs. no-COT results in a table side-by-side to allow for easier comparison.

**Significance:**
It is interesting to study the theoretical bounds on if/how transformers are able to complete different types of problem formulations as this analysis can directly be applied to many real-world LLM problem formulations. It informs humans on the limits of LLM abilities and their failure modes.

**Originality:**
This work builds on an area on a line of work which studied LLM communication complexity. They contribute a new framework (BAPO) for reasoning about communication problems which are bounded by causal attention.

---

> ### Author Rebuttal · Authors · 2025-07-29
>
> Thank you for the positive feedback! It’s helpful to hear how the figures could be made more legible; we’ll work on an easier-to-parse comparison of the CoT and no CoT results. The theoretical bounds on bandwidth do not translate into accuracy bounds, they just separate the easy (top row in Figs 3&4) and hard (bottom row) problems.
>
> To address the questions:
>
> > Can you include the theoretical lower and upper limits of BAPOs with COT similar to what was provided in table 1 for BAPO with no COT?
>
> With enough CoT tokens, the upper bound for bandwidth requirement is (2, 3) for all decidable problems by Theorem 7. We will add this to the table caption.
>
> > You only consider production-grade language models in your empirical analysis. At what point during model training do LLMs become well trained enough to fit the BAPO framework. Presumably, randomly initialized LLMs would not perform well on any task.
>
>  Indeed, models need a certain level of capability to solve even BAPO-easy tasks. As such, we focus on pre-trained LLMs. When capabilities emerge during training is a great question but out of scope for this work.
>
> > I am curious to see if adding in-context examples to the real-world tasks, as is suggested in the discussion (lines 324-330), empirically improves performance?
>
> We don’t expect variable tracking and majority review to be helped by in-context examples. For majority review, we make sure that LLMs can (almost) perfectly classify individual reviews, so adding more examples are unlikely to help. We conjecture that when in-context examples allow for a low-bandwidth shortcut solution (e.g. nearest-neighbor heuristics like we suggest in the discussion) they could improve LLM performance, but that is not the case for the BAPO-hard problems we tested on.

---

> > ### Comment · Reviewer_qUKG · 2025-07-31
> > **Clarifying my question**
> >
> > Thanks for the response. Your responses are reasonable and make sense. However, I think one of my questions was misunderstood:
> > > You only consider production-grade language models in your empirical analysis. At what point during model training do LLMs become well trained enough to fit the BAPO framework. Presumably, randomly initialized LLMs would not perform well on any task.
> >
> > What I meant to ask was: When do you know that the language model you are applying the BAPO framework to is a *production-grade* language model? What distinguishes a *production-grade* language model from non production-grade models.
> >
> > You write "As such, we focus on pre-trained LLMs.", so the natural question emerges: What makes a model pre-trained? How do you know the pretraining corpus is large enough and the length of training was sufficient enough to warrant calling that model "production-grade"? There are thousands of variants of "pre-trained" language models on hugging-face. Are all of them production-grade? Can the BAPO framework be applied to all of them?

---

> > > ### Author Response · Authors · 2025-08-04
> > >
> > > Ah, thanks for the clarification. The BAPO framework is not limited to production-grade LLMs (wherever we choose to draw the line). Our hypothesis is that any transformer-based LLM (e.g., any LLM on hugging-face) will fail on BAPO-hard problems as the input size grows beyond a small constant, regardless of the size/quality of the model. (They may also fail on BAPO-easy problems if they have insufficient capability; in practice, we find that large models like GPT-4o, Claude Haiku, and Gemini Pro all succeed on BAPO-easy problems, even at n = 200. We focus on these "production-grade" models as they show the delineation of BAPO-easy vs BAPO-hard problems in practice, whereas lesser models would just fail on everything.)

---

### Official Review · Reviewer_FeZc · 2025-06-25

**Clarity:** 3
**Significance:** 2
**Originality:** 2
**Rating:** 3
**Confidence:** 3

**Summary:**

This paper delves into the challenges LLMs face when handling tasks that require global reasoning, attributing these difficulties to bandwidth limitations in their internal information transmission. To formalize this issue, the research introduces the Bounded-Attention Prefix Oracle (BAPO) model, which simulates the communication constraints of an LLM's attention heads. This model classifies problems into "BAPO-hard" (requiring high bandwidth) and "BAPO-easy." Experimental results validate that LLMs falter on BAPO-hard tasks. Furthermore, the study demonstrates that Chain of Thought (CoT) can effectively mitigate these bandwidth limitations by decomposing complex tasks into BAPO-easy steps.

**Questions:**

1. How would the BAPO model's theoretical framework and bandwidth requirements change if problems with multi-token, structured outputs were directly modeled, rather than just single-token solutions?
2. How can the hypothesized trade-off between an LLM's "generalization ability" (for natural language tasks) and its "effective bandwidth" (for exact global reasoning tasks) be empirically isolated and quantified?

**Ethical Concerns:**

["NO or VERY MINOR ethics concerns only"]

**Limitations:**

yes

**Quality:**

3

**Strengths And Weaknesses:**

Strengths:

1. Novel Computational Framework (BAPO Model): The paper introduces the Bounded Attention Prefix Oracle (BAPO) model, a new computational framework that formalizes and quantifies communication bandwidth constraints within LLMs' attention heads. This model offers principled explanations for why LLMs struggle with complex global reasoning tasks, abstracting away lower-level transformer details to focus on information flow.
2. Theoretical Explanation for Chain of Thought (CoT) Effectiveness: The research provides a strong theoretical basis for the success of Chain of Thought (CoT), proving that CoT can decompose BAPO-hard problems into a sequence of BAPO-easy steps. It further demonstrates that constant-bandwidth BAPOs become Turing-complete with CoT, theoretically enabling them to solve any decidable problem, albeit potentially requiring a large number of output tokens.
3. Empirical Validation and Practical Implications: The paper's experiments consistently corroborate its theoretical predictions: LLMs perform well on BAPO-easy tasks but systematically fail on BAPO-hard ones, even for relatively small instances across various models like GPT-4o, Claude, and Gemini. This alignment provides practitioners with actionable insights into LLM limitations and suggests mitigation strategies, such as inference-time scaling or hybrid architectures, or even optimizing for low bandwidth requirements during training.

Weaknesses:

1. Simplifying Assumptions in BAPO Model (**Too simple to evaluate the validity**): The BAPO model incorporates several simplifying assumptions that do not fully capture the complexities of real transformer computation. For example, it assumes problems have **single-token** solutions for simplicity, posits unbounded computational power for prefix/suffix streams** (though tempered by the requirement to work for all splits), allows the attention function `g` to base decisions on the **entire suffix** (unlike real causal attention, which communicates across layers), and assumes **perfect positional encoding** and attention only on the token layer. These simplifications might limit its direct applicability to all nuances of LLM behavior.
2. Unidentified Root Cause of Limited Effective Bandwidth: Although the paper hypothesizes that LLM failures stem from limited effective bandwidth, it explicitly states that the "root causes of this severely limited bandwidth" are not fully understood. The authors acknowledge this as a limitation, suggesting it might even be a beneficial feature for generalization rather than a flaw, indicating an incomplete understanding of the underlying mechanism.
3. Loose Theoretical Lower Bounds and Unexplored Problem Space: The paper acknowledges that "many of our lower bounds are loose" and there are "many important problems whose BAPO bandwidths have yet to be explored". This implies that the theoretical characterization of problem hardness in terms of bandwidth is not yet fully refined or comprehensive, leaving gaps in precisely understanding the communication requirements for various computational problems.

---

> ### Author Rebuttal · Authors · 2025-07-29
>
> Thank you for the questions!
>
> > How would the BAPO model's theoretical framework and bandwidth requirements change if problems with multi-token, structured outputs were directly modeled, rather than just single-token solutions?
>
> Once enough output tokens are allowed, this effectively becomes chain of thought, which as we show can reduce bandwidth requirements. And if the number of output tokens is small, then we can theoretically model outputs as a single compound “token” with a larger token vocabulary.
>
> > How can the hypothesized trade-off between an LLM's "generalization ability" (for natural language tasks) and its "effective bandwidth" (for exact global reasoning tasks) be empirically isolated and quantified?
>
> This is a great question, and one that the follow-up experiments suggested by Reviewer mvyW (2nd review) would help answer. If transformers trained to do a task can accomplish it, but LLMs trained for next-token prediction can’t, this indicates that the limitation is due to language modeling rather than architectural capabilities. This will be an interesting direction of future work for the community.
>
> Regarding the listed weaknesses:
>
> - We view the simplicity of the model as a strength: it allows us to explain and isolate a particular failure mechanism without having to account for every detail of the architecture. It also keeps mathematical analysis tractable so we can exactly characterize bandwidth requirements.
> - Identifying the cause of limited bandwidth is a great direction for future work, but we feel that proposing the model, analyzing it theoretically, and testing it empirically is already a significant contribution.
> - Our bandwidth bounds do not need to be tight to be useful; they already provide significant separation between easy and hard problems.

---

> > ### Comment · Reviewer_FeZc · 2025-08-02
> >
> > Thank you for your response, but I still have certain concerns about this modeling approach. Out of caution, I will keep my score unchanged.

---

### Official Review · Reviewer_mvyW · 2025-06-29

**Clarity:** 4
**Significance:** 4
**Originality:** 3
**Rating:** 5
**Confidence:** 3

**Summary:**

In this paper, the authors argue that failures in LLM on tasks requiring global reasoning are due to capacity limits on their internal information flow. They introduce the Bounded Attention Prefix Oracle (BAPO) model, a new computational framework that models these communication bandwidth constraints. The authors use this framework to classify problems as "BAPO-hard" if they require high bandwidth . Their experiments show that major LLMs succeed on BAPO-easy tasks but fail on BAPO-hard ones, corroborating their theoretical predictions.

**Questions:**

Suggestion:
The paper compellingly identifies that a key unanswered question is why modern LLMs, despite their immense scale, exhibit such a limited effective bandwidth. I suggest training small transformer models from scratch specifically on BAPO-hard problems. By systematically varying the models' depth and width during this process, it might be possible to isolate the specific architectural factors that contribute to these communication limits and gain crucial insights into this fundamental issue.

**Ethical Concerns:**

["NO or VERY MINOR ethics concerns only"]

**Final Justification:**

The single weakness I initially noted—that the BAPO model is a significant abstraction of a real transformer—was intended more as a point of clarification on the scope of the model rather than a fundamental flaw. The authors acknowledged this limitation and framed the investigation of specific architectural components as a promising direction for future work inspired by their framework. I find this response to be both appropriate and constructive.
In summary, this paper makes a strong and well-supported contribution to our understanding of LLM limitations. I have no unresolved issues and believe the work is ready for publication.

**Limitations:**

Yes

**Quality:**

4

**Strengths And Weaknesses:**

Strengths:
1. The development of a novel and principled theoretical framework (BAPO) is a valuable contribution, and the BAPO-hardness classification serves as a practical diagnostic tool.
2. The paper successfully bridges the gap between theory and practice by aligning with the actual performance of state-of-the-art models.

Weaknesses:
1. The BAPO model is a significant abstraction of a real transformer as the authors acknowledge.

---

> ### Author Rebuttal · Authors · 2025-07-29
>
> We appreciate the positive feedback! Regarding the suggestion to train small transformers on BAPO-hard tasks to isolate which architectural components contribute to limited bandwidth, this is a great idea and exactly the type of research we hope our paper will inspire. For additional discussion about the simplifying assumptions in the model, please see our responses to reviewers 21nW and FeZc (1st and 3rd reviews).

---

### Official Review · Reviewer_21nW · 2025-07-03

**Clarity:** 4
**Significance:** 3
**Originality:** 2
**Rating:** 5
**Confidence:** 3

**Summary:**

This work introduces BAPO, a framework to reason about when LLMs can solve decision problems. BAPO is centered around communication bandwidth which is used to differentiate between BAPO easy and hard problems. The BAPO framework is able to predict if Transformers will successfully solve problems based on if they require constant or super-constant bandwidths.

**Questions:**

An experiment that I think would be very interesting: Consider the problem of finding the kth larger element in an array and the problem of sorting. If I understand correctly, the former is BAPO-easy (for constant k) while the latter if BAPO-hard. How well do you perform on these problems for different values of k, and different sequence lengths. Do the LLMs successfully solve this problem for different values of k?

**Ethical Concerns:**

["NO or VERY MINOR ethics concerns only"]

**Final Justification:**

I've increased my score because the authors did a great job of addressing some concerns in the rebuttal. In particular, the authors justified why tasks seems to have small communication bandwidths, why a limited set of tasks were covered and ran a couple quick experiments. After the rebuttal, I am a lot more excited about the papers contribution.

**Limitations:**

Yes, the authors have clearly listed out assumptions but could expand a bit more on how these assumptions directly translate to limitations on their theory.

**Paper Formatting Concerns:**

No concerns

**Quality:**

3

**Strengths And Weaknesses:**

**Strengths.** I find the framework to be simple yet elegant for predicting how Transformers perform on certain decision problems. While the idea of using communication complexity / bandwidth has been explored in prior works, the new contribution of this work is to split up the input into a prefix and suffix in order to reason about decision problems.

**Weaknesses.** While the authors have presented the results in 6 different problems, I think this set is very limited. For example, it would be interesting to see if the predictions still hold if we consider problems where the communications bandwidth isn't just 0 or 1. Also, some of the results of CoT also don't seem to hold in practice, i.e., experiments in figure 4 do not show a significant benefit from using chain-of-thought.

I worry that some of the assumption are too strong to clearly make predictions about problems beyond the ones considered in this work. By assuming that the prefix and suffix have infinite computational power, the predictive power of this theory may be a lot weaker.

---

> ### Author Rebuttal · Authors · 2025-07-29
>
> Thank you for the good questions! Please see our responses below.
>
> > some of the results of CoT also don't seem to hold in practice, i.e., experiments in figure 4 do not show a significant benefit from using chain-of-thought
>
> Our CoT theorem says that some low-bandwidth solution exists using sufficiently many reasoning tokens. This is consistent with the results in Figure 4: the low performance of CoT for models aside from o3 and Gemini 2.5 Flash is likely due to the soft limit of 250 words we set, which appears to be insufficient for $n > 50$. Meanwhile, o3 and Gemini 2.5 Flash, without this limit of 250 words (since their reasoning is internal), do show excellent performance on BAPO-hard problems. They do this by using far more reasoning tokens, in some cases 10k+ (Figure 9). Overall, these experiments confirm that CoT can allow models to solve BAPO-hard problems, but that they are not guaranteed to follow a correct CoT procedure and may require a large number of reasoning tokens.
>
> > it would be interesting to see if the predictions still hold if we consider problems where the communications bandwidth isn't just 0 or 1
>
> In trying out various problems, we found that natural BAPO-easy problems tend to have very small constants for their bandwidth requirements. The largest constants we have encountered are (2, 3) for simulating a Turing machine step (Theorem 7), which does correctly predict that CoT can let LLMs solve BAPO-hard problems (as o3 did). This is analogous to the observation that most problems we encounter in P tend to have algorithms whose runtime is a very low-degree polynomial (e.g., we often see algorithms with runtime $n^2$ or $n^3$, but not $n^{15}$). It would be possible to construct problems with higher bandwidth requirements (e.g., rather than Equality, we could consider HammingDistanceAtMostk, which can be solved by a (0, k)-BAPO), and this is a nice suggestion!
>
> > While the authors have presented the results in 6 different problems, I think this set is very limited
>
> As hardness proofs are very challenging, it is common for papers in this area to only consider a small number of problems. For instance, [16] only provides results for two problems (parity and 2Dyck) and the NeurIPS paper [6] analyzes only four problems and presents experiments on two (Index and Dyck). Another closely related NeurIPS paper [30] only considers Match2, Match3, and q-sparse averaging. These papers are still important and valuable, as so little is understood about transformer capabilities that even limited theory is enlightening. Judging by comparable papers, we feel that our problem set is sufficiently broad to be of significant use to the community. This is especially true since we include analysis and results for graph reachability, a broadly applicable problem that naturally connects to deductive reasoning and planning.
>
> > I worry that some of the assumption are too strong to clearly make predictions about problems beyond the ones considered in this work. By assuming that the prefix and suffix have infinite computational power, the predictive power of this theory may be a lot weaker.
>
> We'd like to clarify that our theory predicts only that if a task isn't BAPO-easy, LLMs will fail as input size grows due to limited bandwidth. In practice, LLM failures can happen due to many other factors beyond limited bandwidth, such as limited prefix/suffix compute. By making no assumption about compute, our BAPO framework elegantly isolates this inability to track a large amount of information as a particular mechanism for problem-solving failures. (Since submitting the paper, we have begun using an analogy to “working memory.”) This provides some useful insight: based on this understanding, it’s unlikely that simply scaling the MLPs in a transformer or adding more layers will improve these failings (equivalent to adding “more compute”). Rather, the issue occurs when residual streams communicate with each other through attention (as this is where information tracking and combining occurs).
>
> > Do the LLMs successfully solve [kth largest and sorting] for different values of k?
>
> We ran some additional experiments to check these problems. LLMs have no problem with maximum (k = 1), but struggle for larger k as the input grows, even k = 5 or 10. They also struggle with sorting when the list to be sorted has duplicate items. We believe that k-th largest (for constant k) is BAPO-$\Sigma$-hard, as the number of bits required to communicate the k largest elements in the prefix scales with $|\Sigma|$. We also conjecture that sorting is BAPO-hard, although depending on the input distribution (e.g., with no duplicates when sorting contiguous numbers) there may be shortcut solutions that models can employ in practice.

---

> > ### Comment · Reviewer_21nW · 2025-08-02
> > **Thank you for the rebuttal!**
> >
> > Thank you for the wonderful rebuttal! I will increase my score at the end of the discussion period.
> >
> > Thanks giving a more detailed explanation for Figure 6. While I think the evaluation on 6 tasks is limited, I also agree that it is the case in many other papers and that it is difficult to generate a representative set of synthetic tasks that mirror natural language. I think it would be nice to incorporate some of these points in the main paper.

---

> > > ### Author Response · Authors · 2025-08-08
> > >
> > > Glad to hear we could address your concerns! Yes, we will make updates to the paper clarifying the points raised in reviews, as other readers are likely to have the same question.

---

### Decision · Program_Chairs · 2025-09-17

**Decision:**

Accept (spotlight)

**Comment:**

The paper introduces the Bounded Attention Prefix Oracle (BAPO) model, a simple but expressive abstraction that captures bandwidth limits in the attention mechanism of transformers. Using this framework, the authors (1) demonstrate that advanced LLMs can succeed on BAPO-easy instances but fail on relatively small BAPO-hard ones, and (2) provide an explanation for the empirical success of CoT prompting.

Overall, reviewers praise the novelty and elegance of the model, the clear theoretical–empirical connection, and the improved explanation of several long-standing LLM failure modes. Meanwhile, they are concerned that: 1) this abstraction may be overly simplistic; 2) the explored task set is small; 3) the lower bounds provided are loose; and 4) certain sections require further clarification and better presentation.

The rebuttal is considered. While reviewers acknowledge the modeling limitations of this framework, they generally agree that the paper makes a clean and valuable theoretical contribution to understanding LLMs. The only negative reviewer does not identify any fatal errors and is okay with acceptance.

Taking these factors into account, the AC recommends acceptance, given the importance of better understanding LLMs and the potential of this work to stimulate follow-up research in both theory and practice.